# UniVoxel: Fast Inverse Rendering by Unified Voxelization of Scene Representation

## Abstract

Typical inverse rendering methods focus on learning implicit neural scene representations by modeling the geometry, materials and illumination separately, which entails significant computations for optimization. In this work we design a Unified Voxelization framework for explicit learning of scene representations, dubbed *UniVoxel*, which allows for efficient modeling of the geometry, materials and illumination, thereby accelerating the inverse rendering substantially. To be specific, we propose to encode a scene into a latent volumetric representation, based on which the geometry, materials and illumination can be readily learned via lightweight neural networks in a unified manner. Particularly, we leverage Spherical Gaussians to represent the incident light radiance, which enables the seamless integration of modeling illumination into the unified voxelization framework. Extensive experiments on multiple benchmarks covering diverse scenes demonstrate that *UniVoxel* boosts the optimization efficiency significantly compared to other inverse rendering methods, reducing the per-scene training time from hours to 18 minutes, while achieving favorable reconstruction quality. Code will be released.

## 1 Introduction

Inverse rendering is a fundamental problem in computer vision and graphics, which aims to estimate the scene properties including geometry, materials and illumination of a 3D scene from a set of multi-view 2D images. With the great success of Neural Radiance Fields (NeRF) (Mildenhall et al., 2020) in novel scene synthesis, it has been adapted to inverse rendering by learning implicit neural representations for scene properties. A prominent example is NeRD (Boss et al., 2021), which models materials as the spatially-varying bi-directional reflectance distribution function (SV-BRDF) using MLP networks. Another typical way of learning implicit representations for inverse rendering (Chen et al., 2022b; Zhang et al., 2021b; 2022) is to first pre-train a NeRF or a surface-based model like IDR (Yariv et al., 2020) or NeuS (Wang et al., 2021) to extract the scene geometry, and then they estimate the materials as well as the illumination by learning implicit neural representations for the obtained surface points. A crucial limitation of such implicit learning methods is that they seek to model each individual scene property by learning a complicated mapping function from spatial locations to the property values, which entails significant computations since modeling of each property demands learning a deep MLP network with sufficient modeling capacity. As a result, these methods suffer from low optimization efficiency, typically requiring several hours or even days of training time for each scene, which limits their practical applications.

It has been shown that modeling scenes with explicit representations (Chen et al., 2022a; Fridovich-Keil et al., 2022; Sun et al., 2022) rather than implicit ones is an effective way of accelerating the optimization of NeRF. TensoIR (Jin et al., 2023) makes the first attempt at explicit learning for inverse rendering, which extends TensoRF (Chen et al., 2022a) and performs VM decomposition to factorize 3D spatially-varying scenes features into tensor components. While TensoIR accelerates the optimization substantially compared to the implicit learning methods, it follows the typical way (Chen et al., 2022b; Munkberg et al., 2022; Wu et al., 2023; Zhang et al., 2021b) to model the illumination by learning environment maps which results in two important limitations. First, the methods based on environment maps have to simulate the lighting visibility and indirect lighting for each incident direction of a surface point, which still incurs heavy computational burden. Second, it is challenging for these methods to deal with complex illumination in real-world scenarios due to the limited modeling capability of the environment maps.

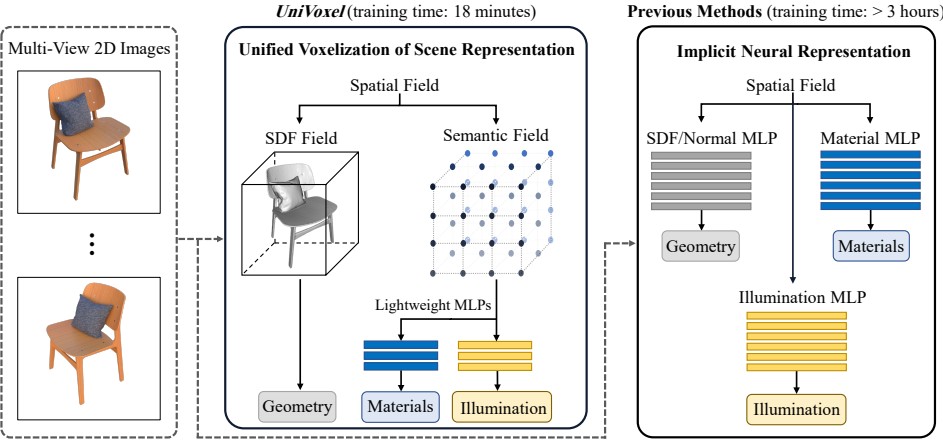

Figure 1: Overview of the proposed *UniVoxel*. Typical methods (Chen et al., 2022b; Zhang et al., 2021b; 2022) for inverse rendering learn implicit neural scene representations from spatial field by modeling the geometry, materials and illumination individually employing deep MLP networks. In contrast, our *UniVoxel* learns explicit scene representations by performing voxelization towards two essential scene elements: SDF field and semantic field, based on which the geometry, materials and illumination can be readily learned with lightweight networks in a unified manner, boosting the optimization efficiency of inverse rendering substantially.

In this work, we propose to boost the optimization efficiency of inverse rendering by explicit voxelization of scene representation. As shown in Figure 1, we devise a Unified Voxelization framework for scene representation, dubbed *UniVoxel*, which encodes a scene into latent volumetric representations consisting of two essential components: 1) Signed Distance Function (SDF) field for capturing the scene geometry and 2) semantic field for characterizing the materials and illumination of the scene. As a result, our *UniVoxel* is able to estimate the materials and illumination of a scene based on the voxelization of the semantic field by learning lightweight MLP networks in a unified manner. Thus, our *UniVoxel* is able to perform inverse rendering more efficiently than other methods, reducing the optimizing time from several hours to 18 minutes.

A crucial challenge of performing inverse rendering with explicit representation lies in the modeling of illumination. Previous methods typically represent the illumination as environment maps, resulting in significant computational cost as they require multi-bounce ray tracing. In this work, we propose a unified illumination modeling mechanism, which leverages Spherical Gaussians (SG) to represent the local incident light radiance. In particular, we model the SG parameters by a unified learning manner with the modeling of geometry and materials, i.e., learning them from the voxelization of the semantic field by a lightweight MLP, which enables seamless integration of illumination modeling into the unified voxelization framework of our *UniVoxel*. Then we can efficiently query the incident light radiance from any direction at any position in the scene. The proposed illumination representation is capable of modelling direct lighting, indirect illumination and light visibility jointly without the need of multi-bounce ray tracing, greatly improving training efficiency.

To conclude, we make the following contributions:

• We design a unified voxelization framework of scene representation, which allows for efficient learning of all essential scene properties for inverse rendering in a unified manner, including the geometry, materials and illumination.

• We propose to model the incident light field with Spherical Gaussians, which enables unified modeling of the illumination with other scene properties based on the learned voxelization of scene representation by our *UniVoxel*.

• Extensive experiments on various benchmarks show that our method achieves favorable reconstruction quality compared to other state-of-the-art approaches for inverse rendering while boosting the optimization efficiency significantly: 40× faster than MII (Zhang et al., 2022) and over 12× faster than Nvdiffrec-mc (Hasselgren et al., 2022).

## 2 RELATED WORK

### 2.1 INVERSE RENDERING

Inverse rendering aims to reconstruct geometry, materials and illumination of the scene from observed images. Early works (Bi et al., 2020b; Chen et al., 2019; 2021; Liu et al., 2019; Nam et al., 2018; Xia et al., 2016) perform inverse rendering with a given triangular mesh as the fixed or initialized scene geometry representation. In contrast, Nvdiffrec (Munkberg et al., 2022) represents scene geometry as triangular mesh and jointly optimize geometry, materials and illumination by a well-designed differentiable rendering paradigm. Nvdiffrec-mc (Hasselgren et al., 2022) further incorporates ray tracing and Monte Carlo integration to improve reconstruction quality. Inspired by the success of NeRF (Mildenhall et al., 2020), some methods (Bi et al., 2020a; Srinivasan et al., 2021) utilize Neural Reflectance Fields to model the scene properties. PhySG (Zhang et al., 2021a) employs Spherical Gaussians to model environment maps. NMF (Mai et al., 2023) devises an optimizable microfacet material model. NeRFactor (Zhang et al., 2021b), L-Tracing (Chen et al., 2022b) and MII (Zhang et al., 2022) adopts the multi-stage framework to decompose the scene under complex unknown illumination. Some recent works apply inverse rendering to more challenging scenarios, such as photometric stereo (Yang et al., 2022), scattering object (Zhang et al., 2023b) and urban scenes (Wang et al., 2023). Although achieving promising results, most of these works require several hours or even days to train for each scene, which limits their practical applications.

### 2.2 EXPLICIT REPRESENTATION

Learning implicit neural representations for scenes with MLP networks typically introduces substantial computation, leading to slow training and rendering. To address this limitation, explicit representation (Fridovich-Keil et al., 2022) and hybrid representation (Chen et al., 2022a; Fang et al., 2022; Liu et al., 2022; Sun et al., 2022) have been explored to model the radiance field for a scene. DVGO (Sun et al., 2022) employs dense voxel grids and a shallow MLP to model the radiance field. TensoRF (Chen et al., 2022a) proposes VM decomposition to factorize 3D spatially-varying scene features to compact low-rank tensor components. Voxurf (Wu et al., 2022) combines DVGO (Sun et al., 2022) and NeuS (Wang et al., 2021) to achieve efficient surface reconstruction. The methods mentioned above are all used for the explicit representation of radiance field in static or dynamic scenes, but cannot be directly applied to inverse rendering task which requires explicit representation of geometry, materials and illumination simultaneously.

There is limited research on explicit representation for inverse rendering. Neural-PBIR (Sun et al., 2023) pre-computes lighting visibility and distill physics-based materials from the radiance field. TensoIR (Jin et al., 2023) extends TensoRF (Chen et al., 2022a) to inverse rendering. However, it does not model the illumination based on the learned explicit representations, but follows the traditional way (Chen et al., 2022b) to represent the illumination as environment maps, which incurs heavy computational cost for simulating lighting visibility and indirect lighting. In this paper, we devise a unified voxelization framework for efficient modeling of the geometry, materials and illumination in a unified manner, reducing the per-scene optimization time to 18 minutes.

## 3 METHOD

### 3.1 OVERVIEW

We devise a Unified Voxelization framework (*UniVoxel*) for explicit scene representation learning, which allows for efficient learning of essential scene properties including geometry, materials and illumination in a unified manner, thereby improving the optimization efficiency of inverse rendering significantly. Figure 2 illustrates the overall framework of our *UniVoxel*. It encodes a scene by performing voxelization toward two essential scene elements: 1) Signed Distance Function (SDF) field for capturing the geometry and 2) semantic field for characterizing the materials and illumination. We model both of them as learnable embeddings for each voxel. As a result, we can obtain the SDF value and semantic feature for an arbitrary position in 3D space by trilinear interpolation efficiently.

For a sampled point along a camera ray for rendering, our *UniVoxel* estimates the albedo, roughness and illumination based on the voxelization of the semantic field by learning quite lightweight MLP

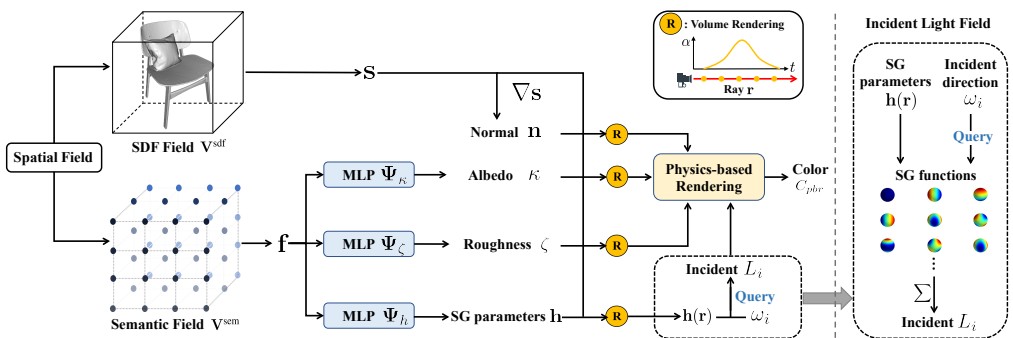

Figure 2: Overall framework of the proposed *UniVoxel*. It performs voxelization towards the SDF field and semantic field to obtain explicit scene representations. The learned volumetric SDF field focuses on capturing the scene geometry while the semantic field characterizes the materials and illumination for the scene. As a result, our *UniVoxel* is able to learn the materials (including the albedo and roughness) and illumination using lightweight MLP networks based on the voxelization of the semantic field. Meanwhile, the surface normal and opacity for an arbitrary 3D point can be easily derived from the voxelization of the SDF field. Hence, our model is able to learn all these scene properties efficiently in a unified manner. In particular, we leverage Spherical Gaussians (SG) to model the incident light field, which allows for unified learning of the illumination with other scene properties based on the voxelization of the scene representation.

networks. Meanwhile, the surface normal and opacity of the sampled point can be easily derived from the voxelization of the SDF field. Leveraging these obtained scene properties, our *UniVoxel* performs volumetric physics-based rendering to reconstruct the 2D appearance of the scene.

## 3.2 PHYSICS-BASED RENDERING

Our model renders a 3D scene into 2D images by applying the classical physics-based rendering formulation (Kajiya, 1986). Formally, for a surface point $\mathbf{x} \in \mathbb{R}^3$, we calculate the outgoing radiance, namely the rendered color $C(\mathbf{x}, \omega_o)$ in 2D, in direction $\omega_o$ as follows:

$$C(\mathbf{x}, \omega_o) = \int_\Omega L_i(\mathbf{x}, \omega_i) \, f_r(\mathbf{x}, \omega_i, \omega_o) \, (\omega_i \cdot \mathbf{n}(\mathbf{x})) d\omega_i, \tag{1}$$

where $\mathbf{n}(\mathbf{x})$ is the surface normal at $\mathbf{x}$ and $L_i(\mathbf{x}, \omega_i)$ denotes the incident light radiance in direction $\omega_i$. $\Omega$ denotes the hemisphere satisfing $\{\omega_i : \omega_i \cdot \mathbf{n}(\mathbf{x}) > 0\}$, while $f_r$ is the BRDF describing the materials at the surface point $\mathbf{x}$. In this work, we adopt the Simplified Disney BRDF model (Burley & Studios, 2012) which derives BRDF from the spatially-varying diffuse albedo $\kappa(\mathbf{x})$ and roughness $\zeta(\mathbf{x})$.

Unlike the typical methods for inverse rendering that estimate the scene properties in Equation 1 including the geometry, materials and illumination based on implicit neural representation learning, our *UniVoxel* obtains these properties by volumetric rendering along camera rays based on the voxelization of scene representation. Specifically, given a camera ray $\mathbf{r}$ with origin $\mathbf{o}$, direction $\mathbf{d}$ and $P$ sampled points $\{\mathbf{x}_i = \mathbf{o} + t_i \mathbf{d} | i = 1, ..., P\}$, we follow NeuS (Wang et al., 2021) to represent the geometry as a zero-level set based on the learned voxelization of the SDF field, and calculate the opacity value $\alpha_i$ at point $\mathbf{x}_i$ by:

$$\alpha_i = \max\left(\frac{\sigma(\mathbf{s}(\mathbf{x}_i)) - \sigma(\mathbf{s}(\mathbf{x}_{i+1}))}{\sigma(\mathbf{s}(\mathbf{x}_i))}, 0\right), \quad \sigma(\mathbf{s}(\mathbf{x}_i)) = (1 + e^{-d\mathbf{s}(\mathbf{x}_i)})^{-1}, \tag{2}$$

where $\mathbf{s}(\mathbf{x}_i)$ is the signed distance at $\mathbf{x}_i$ and $\frac{1}{d}$ is the standard deviation of $\sigma(\mathbf{s}(\mathbf{x}_i))$. Then we compute the albedo $\kappa(\mathbf{r})$ along the ray $\mathbf{r}$ by volume rendering (Mildenhall et al., 2020) as:

$$\kappa(\mathbf{r}) = \sum_{i=1}^{P} T_i \alpha_i \kappa_i, \tag{3}$$

where $T_i = \prod_{j=1}^{i-1}(1 - \alpha_j)$ denotes the accumulated transmittance. We can obtain the roughness $\zeta(\mathbf{r})$ and surface normal $\mathbf{n}(\mathbf{r})$ in the same way. Thus, the essential of such modeling boils down

to learning the geometry and materials for sampled points from the voxelization of the SDF and semantic fields, which is elaborated in Section 3.3. Besides, we will also explicate how to derive the incident light radiance $L_i(\mathbf{x}, \omega_i)$ in Equation 1 from the voxelization of the semantic field in Section 3.4.

## 3.3 Unified Voxelization of Scene Representation

Our *UniVoxel* constructs a unified voxelization framework for explicit learning of scene representations, which allows for efficient estimation of scene properties and fast inverse rendering. To be specific, our *UniVoxel* performs voxelization toward the SDF and semantic fields separately to capture different scene properties. The SDF field focuses on capturing scene geometry while the semantic field characterizes scene materials and illumination. Formally, we learn volumetric embeddings for both of them: $\mathbf{V}^{\mathrm{sdf}} \in \mathbb{R}^{1 \times N_x \times N_y \times N_z}$ for the SDF field and $\mathbf{V}^{\mathrm{sem}} \in \mathbb{R}^{C \times N_x \times N_y \times N_z}$ for the semantic field, where $N_x$, $N_y$ and $N_z$ denote the resolution of voxelization and $C$ is the feature dimension of semantics. The SDF value $\mathbf{s}(\mathbf{x})$ and semantic features $\mathbf{f}(\mathbf{x})$ for an arbitrary position $\mathbf{x} \in \mathbb{R}^3$ in the 3D space can be queried by trilinear interpolation $\mathcal{F}_{\mathrm{interp}}$ on its eight neighboring voxels:

$$\mathbf{s}(\mathbf{x}) = \mathcal{F}_{\mathrm{interp}}(\mathbf{x}, \mathbf{V}^{\mathrm{sdf}}), \quad \mathbf{f}(\mathbf{x}) = \mathcal{F}_{\mathrm{interp}}(\mathbf{x}, \mathbf{V}^{\mathrm{sem}}). \tag{4}$$

The surface normal at position $\mathbf{x}$ can be easily derived based on the learned SDF field of the neighboring samples. For example, we approximate the $x$-component of the surface normal of $\mathbf{x}$ as:

$$\mathbf{n}_x(\mathbf{x}) = (\mathbf{s}(\mathbf{x} + [v, 0, 0]) - \mathbf{s}(\mathbf{x} - [v, 0, 0]))/(2v), \tag{5}$$

where $v$ denotes the size of one voxel. $\mathbf{n}_y(\mathbf{x})$ and $\mathbf{n}_z(\mathbf{x})$ can be calculated in the similar way along the dimension $y$ and $z$, respectively.

Based on the learned volumetric semantic field, our *UniVoxel* models the albedo and roughness using two lightweight MLP networks:

$$\kappa(\mathbf{x}) = \mathbf{\Psi}_\kappa(\mathbf{f}(\mathbf{x}), \mathbf{x}), \quad \zeta(\mathbf{x}) = \mathbf{\Psi}_\zeta(\mathbf{f}(\mathbf{x}), \mathbf{x}), \tag{6}$$

where $\kappa(\mathbf{x})$ and $\zeta(\mathbf{x})$ are the learned albedo and roughness at the position $\mathbf{x}$, respectively.

**Comparison with implicit neural representation learning.** Typical implicit neural representation learning methods seek to model each individual scene property from scratch by learning a mapping function from spatial locations to the property values. Taking the learning of albedo as an example: $\kappa(\mathbf{x}) = \mathbf{\Psi}'_\kappa(\mathbf{x})$, where $\mathbf{\Psi}'_\kappa$ denotes the mapping function of the corresponding MLP network. Such a mapping function is more complicated and more difficult to learn than the mapping function $\mathbf{\Psi}_\kappa$ of our *UniVoxel* shown in Equation 6 in that the learned semantic features $\mathbf{f}(x)$ bridge the gap between the spatial field and the albedo field and thus reduce the learning complexity of the mapping function substantially. Therefore, our *UniVoxel* can utilize a more lightweight MLP network with less modeling capacity to model $\mathbf{\Psi}_\kappa$ than other implicit neural representation learning methods.

## 3.4 Illumination Modeling in the Unified Voxelization

We present two feasible ways to model illumination based on the learned voxelization of scene representations. We first follow classical methods (Chen et al., 2022b; Jin et al., 2023; Zhang et al., 2021a;b; 2022) that learn an environment map to model lighting. Then we propose a unified illumination modeling method by leveraging Spherical Guassians to represent the incident light radiance, which enables seamless integration of illumination modeling into the unified voxelization framework of our *UniVoxel*, leading to more efficient optimization.

**Learning the environment map.** A typical way of modeling illumination is to represent lighting as an environment map (Chen et al., 2022b; Jin et al., 2023; Zhang et al., 2021b; 2022; 2021a), assuming that all lights come from an infinitely faraway environment. Different from other methods (Zhang et al., 2021b; 2022) using an MLP network to predict light visibility, we compute it by volumetric integration. To be specific, considering the surface point $\mathbf{x}$ and an incident direction $\omega_i$, the light visibility $\mathbf{v}(\mathbf{x}, \omega_i)$ is calculated as:

$$\mathbf{v}(\mathbf{x}, \omega_i) = 1 - \sum_{i=1}^{N_l} \alpha_i \prod_{j < i} (1 - \alpha_j), \tag{7}$$

where $N_l$ is the number of sampled points along the ray $\mathbf{r}_i = \mathbf{x} + \omega_i$. Benefiting from the efficiency of the voxel-based representation, $\mathbf{v}(\mathbf{x}, \omega_i)$ can be computed in an online manner. However, sampling a larger number of incident lights or considering multi-bounce ray tracing still results in significant computational cost. To alleviate this issue, we propose to utilize the light field with volumetric representation to model incident radiance.

**Unified illumination modeling based on Spherical Gaussians.** Illumination can be also modeled by learning the light field by implicit neural representation (Yao et al., 2022; Zhang et al., 2023a), which employs a MLP network to learn a mapping function taking a 3D position $\mathbf{x}$ and incident direction $\omega_i$ as input, and producing the light field comprising direct lighting, indirect lighting and light visibility. Such implicit modeling way also suffers from the low optimization efficiency since it demands a deep MLP with sufficient modeling capacity to model the complicated mapping function.

In contrast to above implicit neural representation learning of illumination, we propose to leverage Spherical Gaussians (SG) to represent the incident light field based on the learned unified voxelization framework of our *UniVoxel*. Some previous works (Zhang et al., 2021a; 2022) have explored the use of SG to model illumination. However, they directly represent the entire scene's environment map with SG, requiring expensive multi-bounce ray tracing. In contrast, we utilize SG to model the local incident light radiance at various positions in space. Formally, the parameters of a SG lobe is denoted as $\mathbf{h} = \{a \in \mathbb{R}^3, \lambda \in \mathbb{R}, \mu \in \mathbb{S}^2\}$. Given an incident direction $w_i$ at the position $\mathbf{x}$, the incident light radiance can be obtained by querying the SG functions as the sum of SG lobes:

$$L_i(\mathbf{x}, \omega_i) = \sum_{i=0}^{k} a e^{\lambda(\mu \cdot w_i - 1)}, \tag{8}$$

where $k$ denotes the number of SG lobes. Herein, we model the essential component of the SG parameters $\mathbf{h}$ in a unified learning manner with the modeling of the geometry and materials as shown in Section 3.3 based on the voxelization of the scene representation:

$$\mathbf{h}(\mathbf{x}) = \boldsymbol{\Psi}_h(\mathbf{f}(\mathbf{x}), \mathbf{x}), \tag{9}$$

where $\boldsymbol{\Psi}_h$ denotes a lightweight MLP network. Then we obtain the $\mathbf{h}(\mathbf{r})$ along the camera ray $\mathbf{r}$ by the volume rendering shown in Equation 3 with $\kappa_i$ replaced by $\mathbf{h}(\mathbf{x}_i)$. Thus, we can efficiently query incident light radiance from an arbitrary direction at a surface point. As a result, our *UniVoxel* is able to integrate illumination modeling into the constructed unified voxelization framework, which boosts the optimization efficiency substantially.

Note that some prior works (Garon et al., 2019; Rudnev et al., 2022) use Spherical Harmonics (SH) instead of SG to model illumination with the crucial limitation that they fail to recover the high-frequency lighting. We will compare our model with this modeling way in Section 4.3.

**Extension to varying illumination conditions.** Thanks to the flexibility of the proposed illumination model, our *UniVoxel* can be easily extended to varing illumination conditions, where each view of the scene can be captured under different illuminations. Specifically, given $N_v$ multi-view images of a scene, we maintain a learnable view embedding $\mathbf{e} \in \mathbf{R}^{N_v \times C_v}$, where $C_v$ is the dimension of the view embedding. Then we employ the view embedding of current view as the additional input of $\boldsymbol{\Psi}_h$ to predict the SG parameters, so the Equation 9 is modified as:

$$\mathbf{h}(\mathbf{x}) = \boldsymbol{\Psi}_h(\mathbf{f}(\mathbf{x}), \mathbf{x}, \mathbf{e_x}). \tag{10}$$

Thus, our illumination model is able to model the view-varying illumination conditions.

### 3.5 OPTIMIZATION

The proposed *UniVoxel* is optimized with three types of losses in an end-to-end manner.

**Reconstruction loss.** Similar to other inverse rendering methods (Chen et al., 2022b; Zhang et al., 2021b; 2022), we compute the reconstruction loss between the physics-based rendering colors $C_{\text{pbr}}$ and the ground truth colors $C_{\text{gt}}$. To ensure a stable geometry during training, we use an extra radiance field taking features $\mathbf{f}$, position $\mathbf{x}$, and normal $\mathbf{n}$ as inputs to predict colors $C_{\text{rad}}$. Thus, the reconstruction loss is formulated as:

$$\mathcal{L}_{\text{rec}} = \lambda_{\text{pbr}} \|C_{\text{pbr}} - C_{\text{gt}}\|_2^2 + \lambda_{\text{rad}} \|C_{\text{rad}} - C_{\text{gt}}\|_2^2, \tag{11}$$

Table 1: Quantitative evaluation on the MII synthetic dataset. We report the mean metrics over 200 novel validation views of all 4 scenes. The best result is denoted in bold, while the second best is underlined. Following previous works (Chen et al., 2022b; Munkberg et al., 2022; Zhang et al., 2021b), we align the albedo with the ground truth before calculating the metrics to eliminate the scale ambiguity. The NVS results of our *Univoxel* are generated by physics-based rendering for a fair comparison, although the ones generated by the radiance field are better.

| Method | NVS | | | Albedo | | | Relighting | | | Roughness | Time↓ |
|---|---|---|---|---|---|---|---|---|---|---|---|
| | PSNR↑ | SSIM↑ | LPIPS↓ | PSNR↑ | SSIM↑ | LPIPS↓ | PSNR↑ | SSIM↑ | LPIPS↓ | MSE↓ | |
| NerFactor | 22.795 | 0.917 | 0.151 | 19.486 | 0.864 | 0.206 | 21.537 | 0.875 | 0.171 | - | >2 days |
| MII | 30.727 | 0.952 | 0.085 | 28.279 | 0.935 | 0.072 | 28.674 | 0.950 | 0.091 | 0.008 | 14 hours |
| Nvdiffrec-mc | 34.291 | 0.967 | 0.067 | 29.614 | 0.945 | 0.075 | 24.218 | 0.943 | 0.078 | 0.009 | 4 hours |
| TensoIR | 35.804 | 0.979 | **0.049** | **30.582** | 0.946 | 0.065 | **29.686** | 0.951 | 0.079 | 0.015 | 3 hours |
| *UniVoxel* | **36.232** | **0.980** | **0.049** | 29.933 | **0.957** | **0.057** | 29.445 | **0.960** | **0.070** | **0.007** | 18 minutes |

where $\lambda_{\text{pbr}}$ and $\lambda_{\text{rad}}$ are the loss weights.

**Smoothness constraints.** We apply a smoothness loss to regularize the albedo near the surfaces:

$$\mathcal{L}_{s-\kappa} = \sum_{\mathbf{x}_{surf}} \|\mathbf{\Psi}_\kappa(\mathbf{x}_{surf}) - \mathbf{\Psi}_\kappa(\mathbf{x}_{surf} + \epsilon)\|_2^2. \tag{12}$$

where $\epsilon$ is a random variable sampled from a normal distribution. Similar regularization are conducted for normal $\mathcal{L}_{s-n}$ and roughness $\mathcal{L}_{s-\zeta}$. Thus, the smoothness loss is formulated as:

$$\mathcal{L}_{\text{smo}} = \lambda_\kappa \mathcal{L}_{s-\kappa} + \lambda_\zeta \mathcal{L}_{s-\zeta} + \lambda_n \mathcal{L}_{s-n}, \tag{13}$$

where $\lambda_\kappa$, $\lambda_\zeta$ and $\lambda_n$ are balancing weights for the different terms.

**Illumination regularization.** Neural incident light field could lead to material-lighting ambiguity (Yao et al., 2022) due to the lack of constraints. We propose two regularization constraints to alleviate this ambiguity. First, we encourage a smooth variation of lighting conditions between adjacent surface points by applying a smoothing regularization on the Spherical Gaussian parameters:

$$\mathcal{L}_{\text{sg}} = \sum_{\mathbf{x}_{surf}} \|\mathbf{h}(\mathbf{x}_{surf}) - \mathbf{h}(\mathbf{x}_{surf} + \epsilon)\|_2^2. \tag{14}$$

Since the incident light is primarily composed of direct lighting, which is mostly white lighting (Munkberg et al., 2022), the second regularization of illumination is performed on the incident light by penalizing color shifts:

$$\mathcal{L}_{\text{white}} = |L_i(\mathbf{x}, \omega_i) - \overline{L_i(\mathbf{x}, \omega_i)}|, \tag{15}$$

where $\overline{L_i(\mathbf{x}, \omega_i)}$ denotes the average of the incident intensities along the RGB channel. Thus, the illumination regularization is formulated as:

$$\mathcal{L}_{\text{reg}} = \lambda_{\text{sg}}\mathcal{L}_{\text{sg}} + \lambda_{\text{white}}\mathcal{L}_{\text{white}}, \tag{16}$$

where $\lambda_{\text{sg}}$ and $\lambda_{\text{white}}$ are the weights of regularization loss.

Combing all the losses together, our *UniVoxel* is optimized by minimizing:

$$\mathcal{L} = \mathcal{L}_{\text{rec}} + \mathcal{L}_{\text{smo}} + \mathcal{L}_{\text{reg}}. \tag{17}$$

## 4 EXPERIMENTS

### 4.1 EXPERIMENTAL SETUP

We conduct experiments on both synthetic and real-world datasets for evaluation. First, we select 4 challenging scenes from the MII synthetic dataset (Zhang et al., 2022) for experiments. Each scene consists of 100-200 training images and 200 validation images from novel viewpoints. We show both the quantitative and qualitative results for the reconstructed albedo, roughness, novel view synthesis (NVS) and relighting. Furthermore, we evaluate our approach on 5 scenes of the NeRD real-world dataset (Boss et al., 2021). The implementation details are described in the appendix.

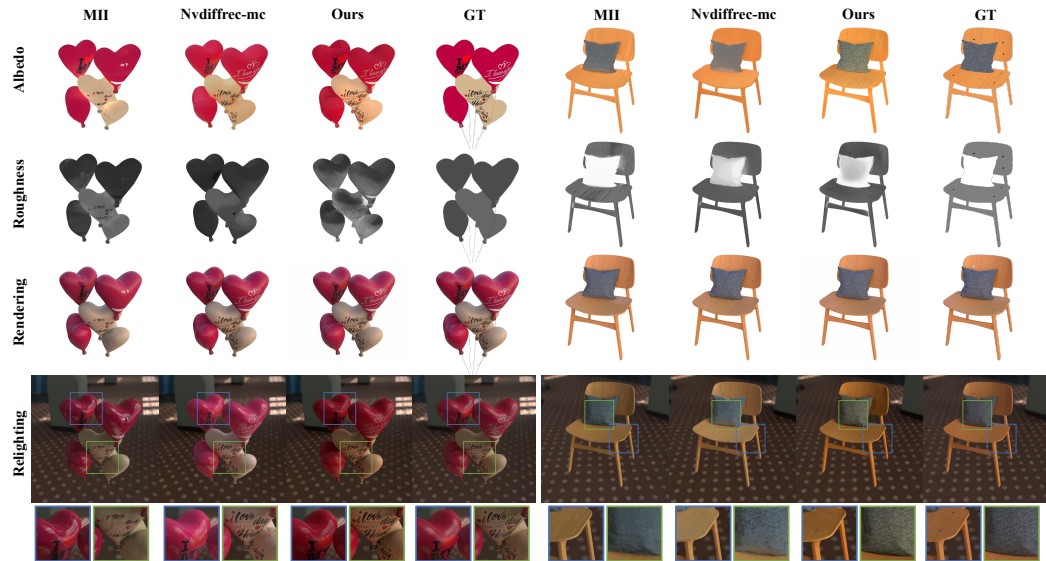

Figure 3: Qualitative comparisons on 2 scenes from the MII synthetic dataset. More qualitative results are shown in Section F of the appendix.

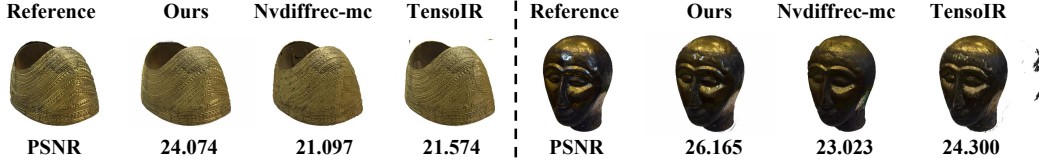

Figure 4: Novel view synthesis results on 2 real-world scenes in a fixed environment from the NeRD dataset: *Ethiopian Head* and *Gold Cape*. We report the average PSNR below each image.

## 4.2 COMPARISONS WITH STATE-OF-THE-ART METHODS

**Results on synthetic datasets.** We compare our *UniVoxel* with NeRFactor (Zhang et al., 2021b), MII (Zhang et al., 2022), Nvdiffrec-mc (Hasselgren et al., 2022) and TensoIR (Jin et al., 2023) on the MII synthetic dataset, adopting Peak Signal-to-Noise Ratio (PSNR), Structural Similarity Index Measure (SSIM), and Learned Perceptual Image Patch Similarity (LPIPS) (Zhang et al., 2018) as the quantitative metrics. As shown in Table 1, our *UniVoxel* outperforms other methods in most metrics while taking much less training time. We show the qualitative results in Figure 3. It can be observed that MII fails to restore the high-frequency details on the albedo maps, such as the text on the airballoons, the nails on the chair and the textures on the pillow. Nvdiffrec-mc performs badly in specular areas. In contrast, our *UniVoxel* can produce accurate reconstructions and relighting.

**Results on real-world datasets.** To demonstrate the generalization ability of our method, we conduct experiments on 5 scenes from the NeRD real-world dataset. First, we evaluate on 2 scenes which are captured in a fixed environment. The qualitative and quantitative results of novel view synthesis are shown in Figure 4. Both Nvdiffrec-mc and TensoIR suffer from various artifacts such as holes on *Gold Cape* and specular areas on *Ethiopian Head*, while our *UniVoxel* achieves better rendering results. Furthermore, we evaluate on the other 3 scenes captured under varying illumination. The qualitative results are shown in Figure 5. Due to the difficulty of estimating the complex illumination in the wild via environment maps, TensoIR fails to recover the geometry and materials of the objects, thus causing poor relighting results. In contrast, our *UniVoxel* produces plausible normal, albedo and roughness, and achieves realistic relighting.

## 4.3 ABLATION STUDIES FOR ILLUMINATION MODELING

To showcase the effectiveness of our proposed voxelization representation of the incident light field, we perform an ablation for illumination modeling. The results are reported in Table 2. In 'Envmap',

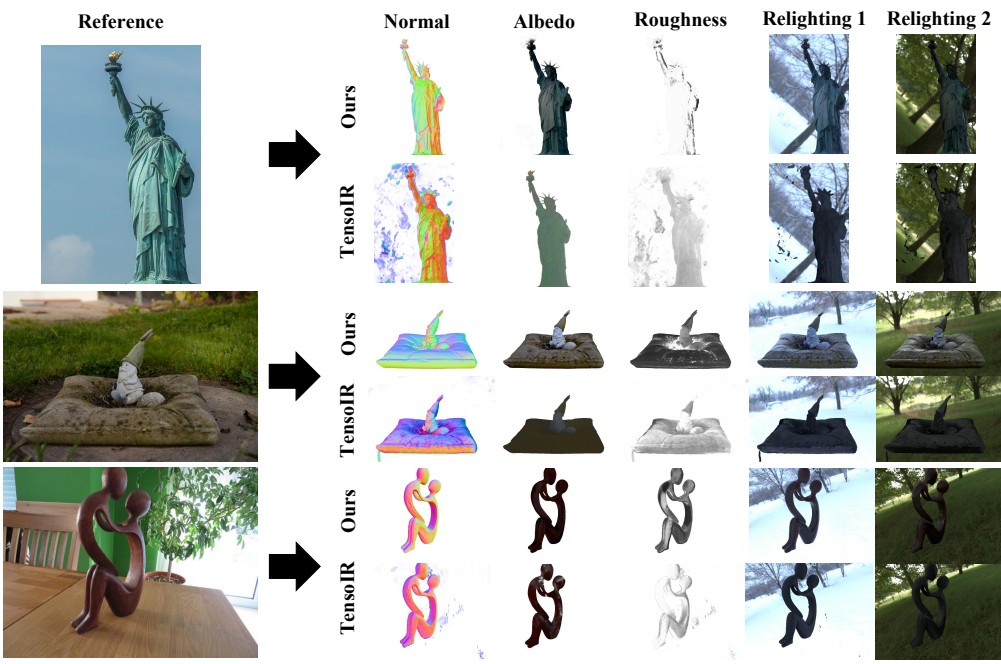

Figure 5: Qualitative comparisons on 3 real-world scenes from the NeRD dataset. All the scenes are captured under varying illumination, which are more challenging. More qualitative results are shown in Section G of the appendix.

Table 2: Ablation studies: comparison of different illumination modeling methods. All methods are built on our unified voxelization framework to have a fair comparison.

| Method | NVS PSNR↑ | Albedo PSNR↑ | Roughness MSE↓ | Relighting PSNR↑ | Time↓ |
|---|---|---|---|---|---|
| Envmap | 34.185 | 27.368 | 0.012 | 27.446 | 58 minutes |
| MLP (NeILF) | **36.355** | 28.974 | **0.007** | 28.694 | 30 minutes |
| SH | 35.328 | 29.185 | 0.020 | 28.981 | 22 minutes |
| SG (Ours) | 36.232 | **29.933** | **0.007** | **29.445** | **18 minutes** |

we represent the illumination as an environment map, parameterized by a mixture of 128 Spherical Gaussians. It is not surprising that the training time is much longer since it requires computing the lighting visibility via Equation 7 for each incident light. And the rendering results are also worse compared to our method. 'MLP(NeILF)' utilizes the neural incident light field to directly predict the incident light using an 8-layer MLP with a feature dimension of 128. Its training time is about twice as long as ours. Besides, without constraints for the light field, the lighting would be baked into the estimated albedo, resulting in decrease in the quality of albedo. In contrast, thanks to the voxelization of the incident light, our *UniVoxel* can easily constrain the lighting conditions in adjacent regions, thereby alleviating the ambiguity between materials and illumination. 'SH' represents the incident light field via Spherical Harmonics (SH) instead of Spherical Gaussians. We employ 3rd-order SH and predict the SH coefficients using the MLP mentioned in Equation 9. However, due to the difficulty of modeling high-frequency lighting with SH, the quality of the generated materials is comparatively poor. The qualitative comparisons of different illumination models and the ablation studies for each loss are shown in Section C of the appendix.

## 5 CONCLUSION

We propose a unified voxelization framework for inverse rendering (*UniVoxel*). It learns explicit voxelization of scene representations, which allows for efficient modeling of all essential scene properties in a unified manner, boosting the inverse rendering significantly. Particularly, we leverage Spherical Gaussians to learn the incident light field, which enables the seamless integration of illumination modeling into the unified voxelization framework. Extensive experiments show that *UniVoxel* outperforms state-of-the-art methods in terms of both quality and efficiency.

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

# APPENDIX

## A    OVERVIEW

We have proposed a novel inverse rendering framework based on the unified voxelization of scene representation. In this appendix, we will present more results of our method. First, we will discribe the implementation details in Section B and present additional ablation studies in Section C. Then we will show additional results on the MII (Zhang et al., 2022) synthetic dataset and the NeRD (Boss et al., 2021) real-wold dataset in Section F and Section G, respectively. Finally, we will discuss the limitation of our method in Section H.

## B    IMPLEMENTATION DETAILS

To calculate the outgoing radiance $C(\mathbf{x}, \omega_o)$ by Equation 1 in the main paper using a finite number of incident lights, we utilize Fibonacci sampling over the half sphere to sample incident lights for each surface point, and the sampling number is set to 128. As for relighting, the incident light field obtained from previous training is not applicable to the new illumination. Therefore, we adopt a similar procedure as the previous methods (Zhang et al., 2021b; Jin et al., 2023), where we compute light visibility using Eq 7 in the main paper and consider only direct lighting.

We adopt the coarse-to-fine training paradigm used in (Sun et al., 2022). During the coarse stage, we only optimize the radiance field branch to accelerate training. The resolution of voxelization is set to $96^3$ in the coarse stage and $160^3$ in the fine stage. Each lightweight MLP network in our *UniVoxel* comprises 3 hidden layers with 192 channels. The number of the feature channels of the semantic field $\mathbf{V}^{\text{sem}}$ is 6. The sampling step size along a ray is set to half of the voxel size. The number of Spherical Gaussian lobes is $k = 16$. The weights of the losses are tuned to be $\lambda_{pbr} = 1.0$, $\lambda_{rad} = 1.0$, $\lambda_n = 0.002$, $\lambda_\kappa = 0.0005$, $\lambda_\zeta = 0.0005$, $\lambda_{sg} = 0.0005$ and $\lambda_{white} = 0.0001$. We use the Adam optimizer (Kingma & Ba, 2014) with a batch size of 8192 rays to optimize the scene representation for 10k iterations in both the coarse and fine stages. The base learning rate is 0.001 for MLP networks and 0.1 for $\mathbf{V}^{\text{sdf}}$ and $\mathbf{V}^{\text{sem}}$. And the learning rate for $\mathbf{V}^{\text{sdf}}$ is reduced to 0.005 in the fine stage. For the experiments on NeRD real-world dataset, we adopt the extended version of our illumination model, and set the dimension of the view embedding to $C_v = 6$.

We run all experiments on a single RTX 3090 GPU, and the training time of other baselines is measured on the same machine.

## C    ADDITIONAL ABLATION STUDIES

### C.1    EFFECTIVENESS OF EACH LOSS

We conduct experiments to explore the effectiveness of each loss used in our method. As shown in Table 3, the performance of our method does not rely on the introduction of the extra radiance field, although it can provide more stability during training and slightly improve the quality of predicted materials. Smoothness constraints and regularization for white light also contribute to enhancing the reconstruction to a certain extent. While the regularization for Spherical Gaussians (SG) leads to a slight decrease in the results of novel view synthesis and roughness, it improves the quality of albedo and yields better visualization. We compare the reconstructed albedo optimized with and without the regularization for Spherical Gaussians in Figure 6. Without $L_{\text{sg}}$, the illumination tends to be baked into the predicted albedo, resulting in poor texture recovery of the pillow, which demonstrates the effectiveness of our proposed regularization in alleviating the material-lighting ambiguity.

### C.2    COMPARISON OF DIFFERENT ILLUMINATION MODELS

We show the qualitative results of different illumination models in Figure 7. Using the environment map to model illumination leads to poor albedo map due to the computational challenges involved in computing light visibility and indirect lighting, making optimization difficult. When employing MLP to predict incident radiance directly, as done by NeILF (Yao et al., 2022), the lighting tends to be baked into the albedo map without constraints for the illumination. Representing incident

lights using Spherical Harmonics (SH) fails to recover high-frequency illumination, causing color deviations in certain regions of the albedo. The visualization aligns with the quantitative results presented in Table 2 of the main paper.

Table 3: Ablation studies of each loss.

| Method | NVS PSNR↑ | Albedo PSNR↑ | Roughness MSE↓ |
|---|---|---|---|
| UniVoxel | 36.232 | 29.933 | 0.007 |
| w/o radiance field | 36.304 | 29.604 | 0.008 |
| w/o smoothness constraints $L_{smo}$ | 36.216 | 29.654 | 0.008 |
| w/o regularization for white lights $L_{white}$ | 36.230 | 29.781 | 0.007 |
| w/o regularization for SG $L_{sg}$ | 36.260 | 29.085 | 0.006 |

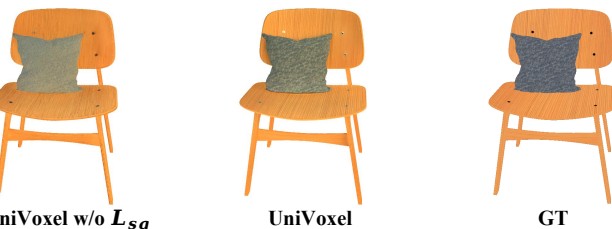

**UniVoxel w/o $L_{sg}$**    **UniVoxel**    **GT**

Figure 6: Visualization of the albedo maps reconstructed by our method with/without the regularization for Spherical Gaussians.

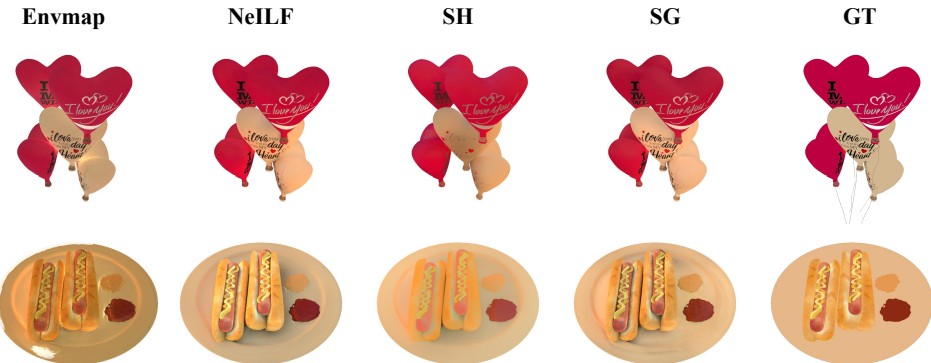

Figure 7: Visualization of the reconstructed albedo maps by different illumination models.

### C.3 VISUALIZATION FOR INCIDENT LIGHTS

We show the incident light maps in Figure 8. Our illumination model is able to represent the effect of direct lighting, occlusions and indirect lighting simultaneously. As shown in the *airballoons* scene of Figure 8, point $x_1$ locates at the top of the balloons, therefore receiving predominantly ambient lights as its incident lights. On the other hand, point $x_2$ is located at the saddle point of the balloons, where the surrounding surfaces exhibit low roughness. Consequently, a portion of the incident lights in its incident light map is composed of red light reflected from the neighboring surfaces. In contrast, The environment maps learned by TensoIR (Jin et al., 2023) only model direct lighting, thus lack the capability to capture such spatially-varying indirect lighting.

## D  RESULTS ON THE SHINY BLENDER DATASET

We conducted experiments on the challenging Shiny Blender dataset (Verbin et al., 2022). As shown in Table 4, our *UniVoxel* achieves better geometric quality compared to other methods. In Fig 9, we visualize the normal maps produced by different methods, and it can be observed that our *UniVoxel*

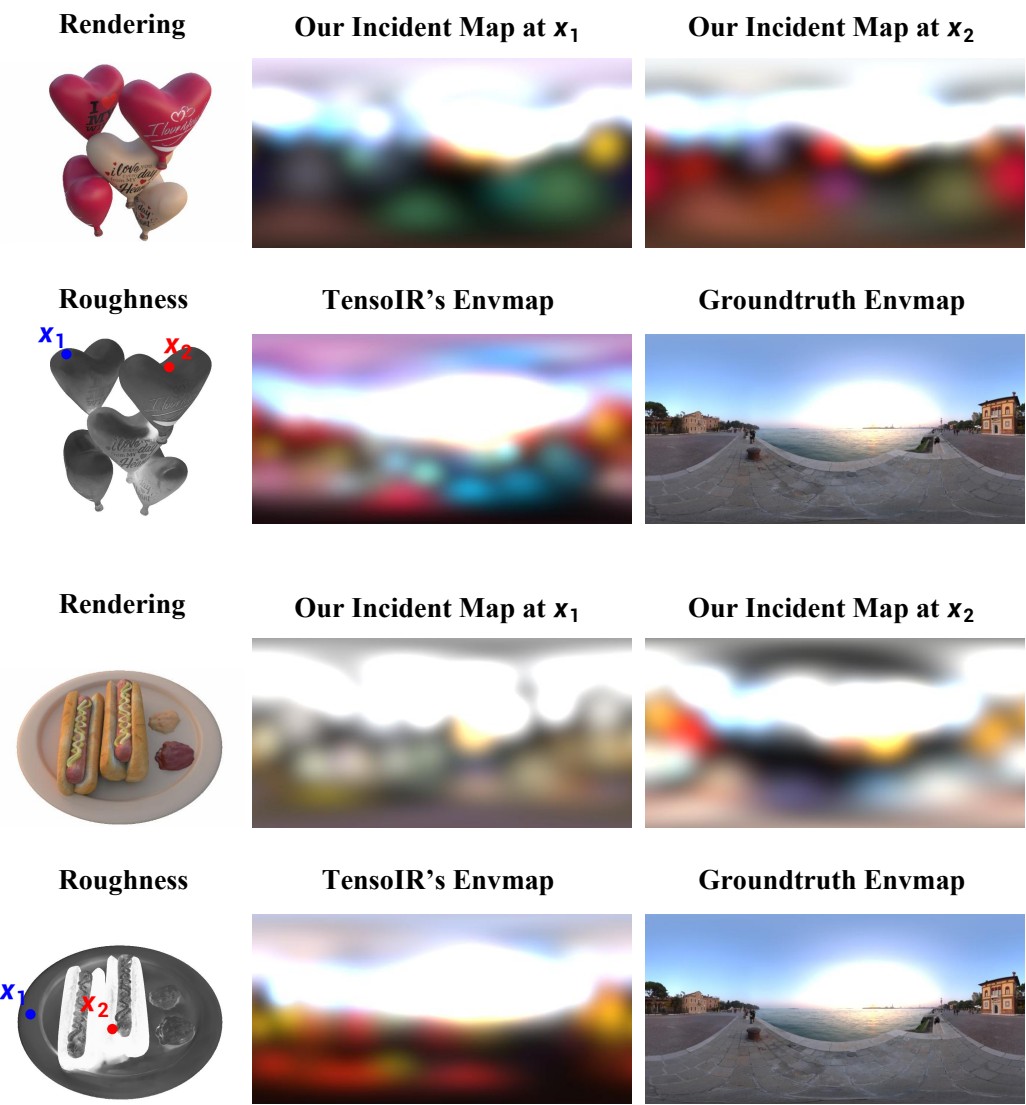

| Rendering | Our Incident Map at $x_1$ | Our Incident Map at $x_2$ |
| Roughness | TensoIR's Envmap | Groundtruth Envmap |
| Rendering | Our Incident Map at $x_1$ | Our Incident Map at $x_2$ |
| Roughness | TensoIR's Envmap | Groundtruth Envmap |

Figure 8: Visualization of the incident light maps reconstructed by our method. Note that our incident light field is designed for modeling both direct lighting and indirect lighting, while the environment map learned by TensoIR is only designed for modeling direct lighting.

Table 4: Quantitative evaluation on the Shiny Blender dataset. We report the per-scene mean angular error (MAE°) of the normal vectors as well as the mean MAE° over scenes.

| MAE° ↓ | teapot | toaster | car | ball | coffee | helmet | mean |
|---|---|---|---|---|---|---|---|
| Mip-NeRF (Barron et al., 2021) | 66.470 | 42.787 | 40.954 | 104.765 | 29.427 | 77.904 | 60.38 |
| Ref-NeRF (Verbin et al., 2022) | 9.234 | 42.870 | 14.927 | 1.548 | 12.240 | 29.484 | 18.38 |
| Voxurf (Wu et al., 2022) | 8.197 | 23.568 | 17.436 | 30.395 | 8.195 | 20.868 | 18.110 |
| TensoIR (Jin et al., 2023) | 8.709 | 60.968 | 35.483 | 100.679 | 15.728 | 76.915 | 49.747 |
| *Univoxel* | 6.855 | 11.515 | 8.987 | 1.635 | 23.654 | 3.108 | **9.292** |

recovers geometry in the specular regions more accurately than TensoIR and Voxurf. Additionally, we present the recovered geometry, materials and illumination in Fig 10. It can be seen that TensoIR fails to reconstruct materials in the specular regions and bakes the lighting into the albedo maps, whereas our method predicts realistic materials.

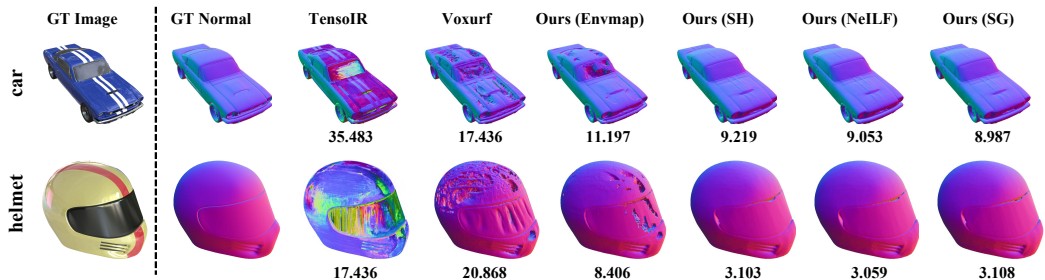

Figure 9: Qualitative comparison of normal maps on 2 scenes from the Shiny Blender dataset. We report the average MAE° below each image.

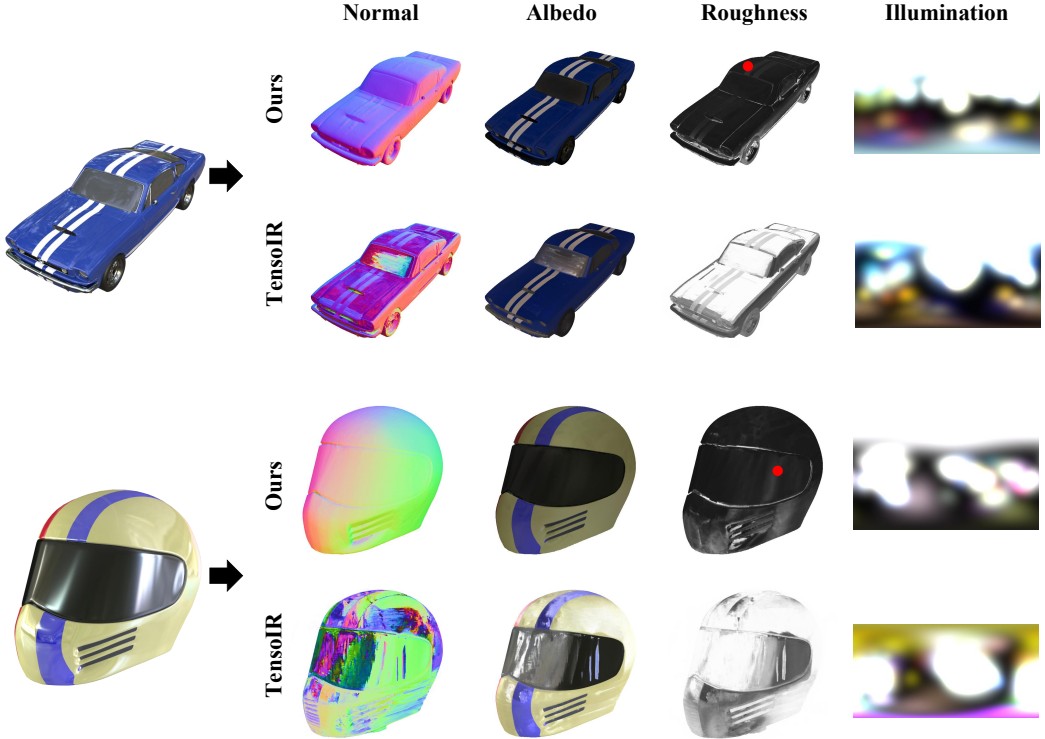

Figure 10: Qualitative comparison of geometry, materials and illumination on 2 scenes from the Shiny Blender dataset. For our method, we generate the incident light maps at the location of the red points in the roughness maps.

# E    EFFECT OF THE SG SMOOTHNESS

we compare the estimated albedo maps with different SG smoothness loss weight $\lambda_{sg}$ of Eq 16 on *StateOfLiberaty* scene from the NeRD dataset in Fig 11. It can be observed that using a larger $\lambda_{sg}$ will result in shadows appearing on the albedo maps. Due to the more complex lighting conditions in outdoor scenes, it is advisable to reduce the constraints on illumination to eliminate these shading components.

# F    ADDITIONAL RESULTS ON THE MII SYNTHETIC DATASET

In Figures 12 to 15, we present complete qualitative results on 4 scenes from the MII synthetic dataset: *air balloons*, *chair*, *hotdog* and *jugs*. Compared to baseline methods, our *UniVoxel* demon-

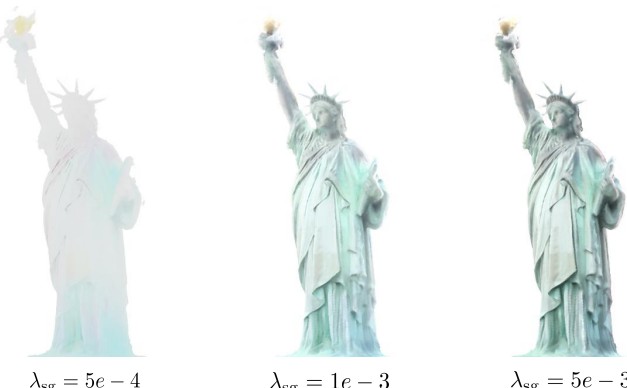

$\lambda_{\text{sg}} = 5e - 4$        $\lambda_{\text{sg}} = 1e - 3$        $\lambda_{\text{sg}} = 5e - 3$

Figure 11: Comparison of albedo with different SG smoothness loss weight.

strates superior reconstruction quality in high-frequency details, which is consistent with the quantitative results presented in Table 1 of the main paper.

## G  ADDITIONAL RESULTS ON THE NERD REAL-WORLD DATASET

In Figures 16 to 18, we show complete qualitative results on the 3 scenes from the NeRD real-word dataset: *StatueOfLiberty*, *Gnome* and *MotherChild*. Although there is no ground truth for reference, we can observe that all baseline methods exhibit poor reconstruction quality in these scenes. The main reason is that the environment maps cannot model the complex lighting conditions in the real world. In contrast, our *UniVoxel* is able to handle various illumination effects, enabling the recovery of geometry and material with relatively superior quality, and the generation of more photo-realistic relighting images.

## H  LIMITATION

Similar to other explicit scene representation methods (Fridovich-Keil et al., 2022; Sun et al., 2022; Wu et al., 2022), our *UniVoxel* requires relatively large amount of memory for storing scene properties, which we intend to investigate in future work.

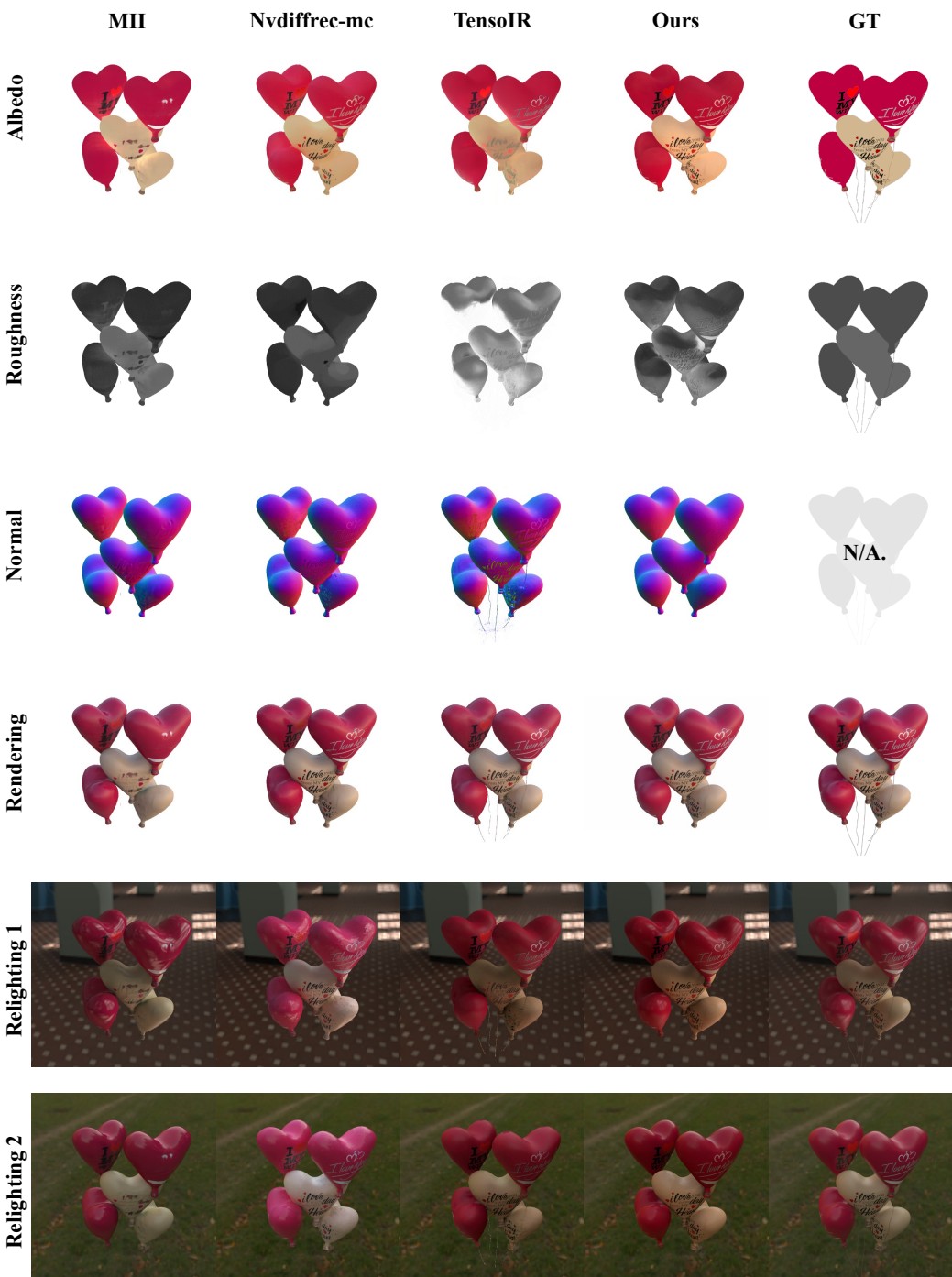

Figure 12: Qualitative comparison on *air balloons* from the MII synthetic dataset.

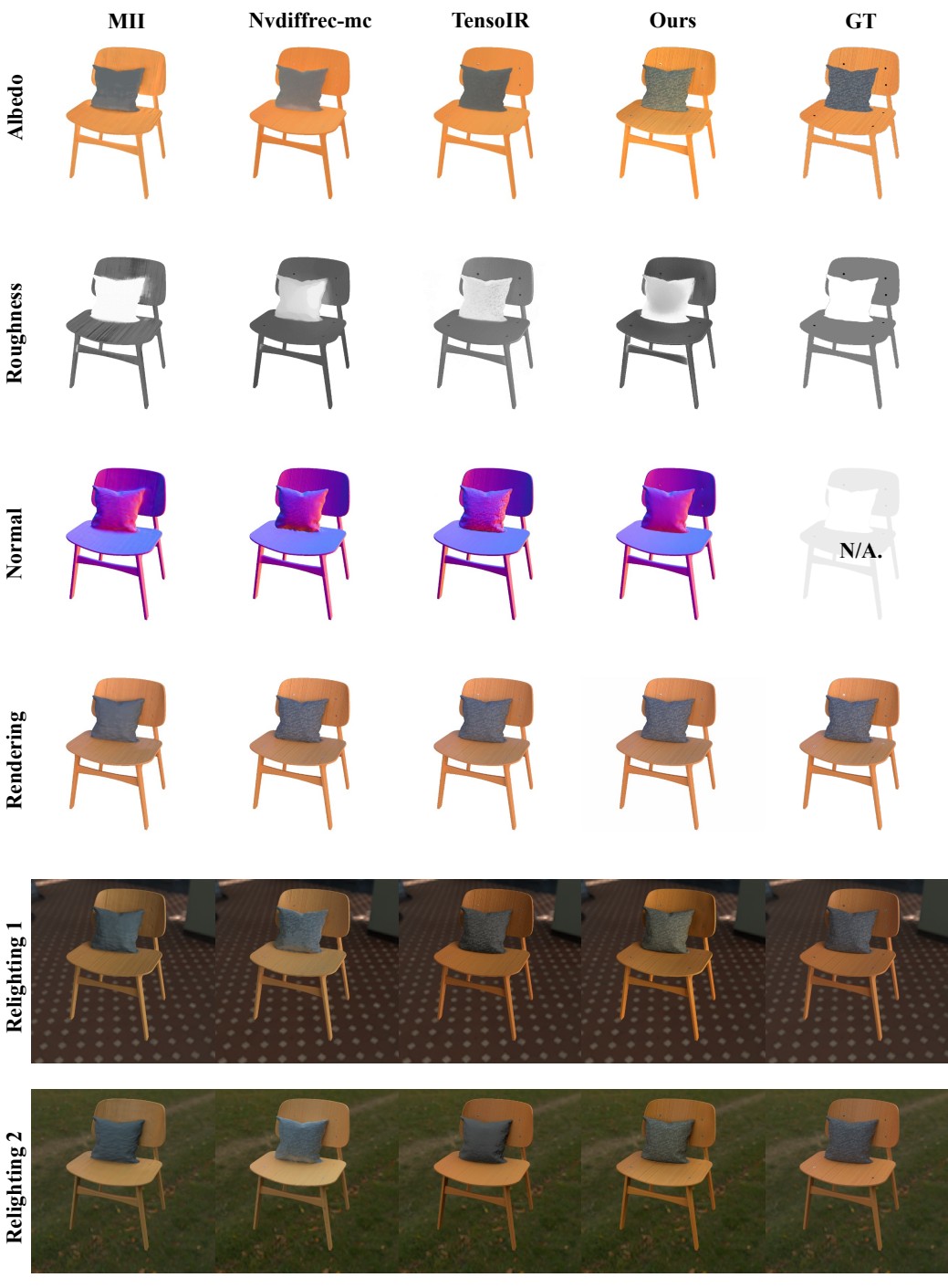

Figure 13: Qualitative comparison on *chair* from the MII synthetic dataset.

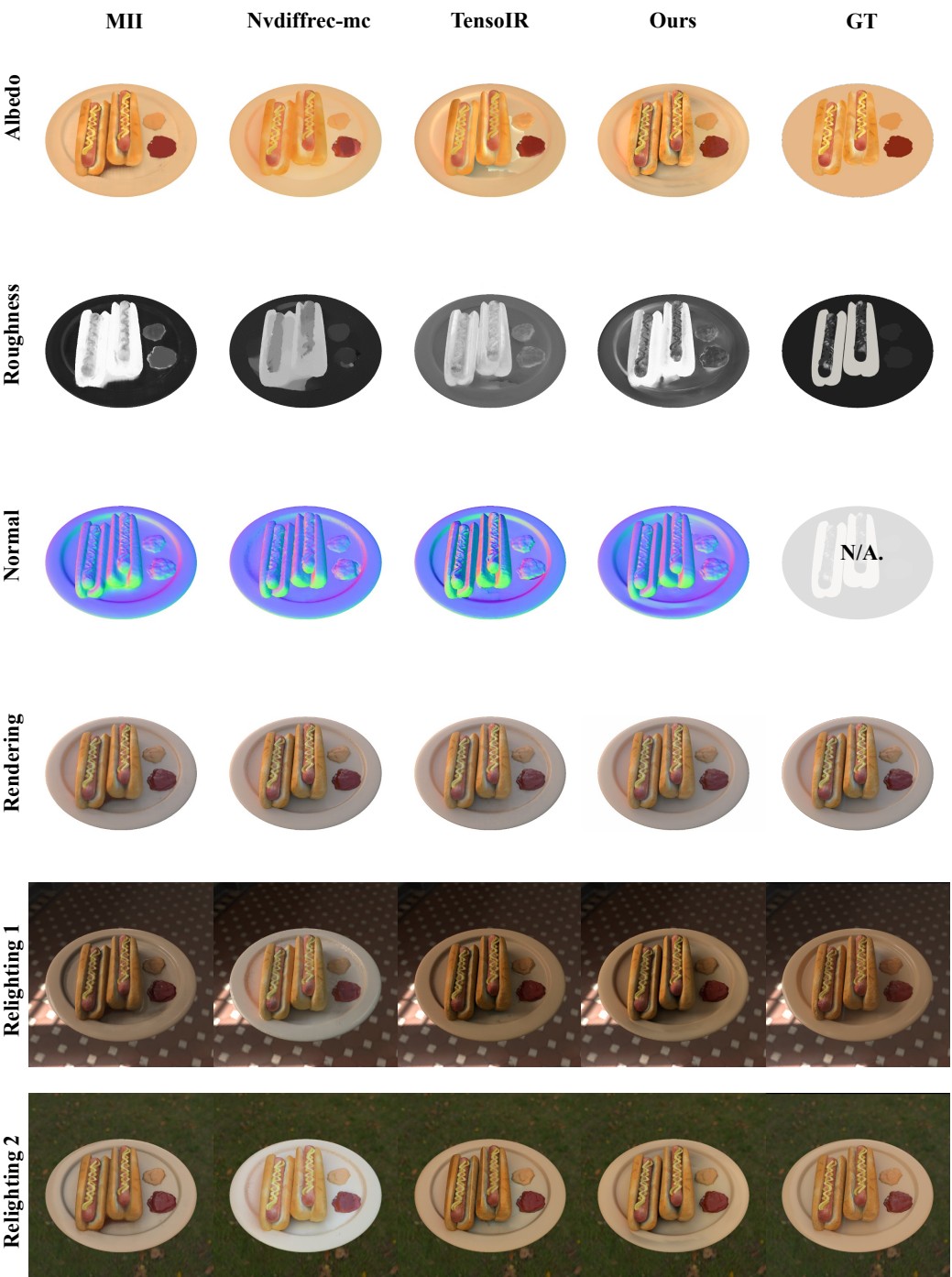

Figure 14: Qualitative comparison on *hotdog* from the MII synthetic dataset.

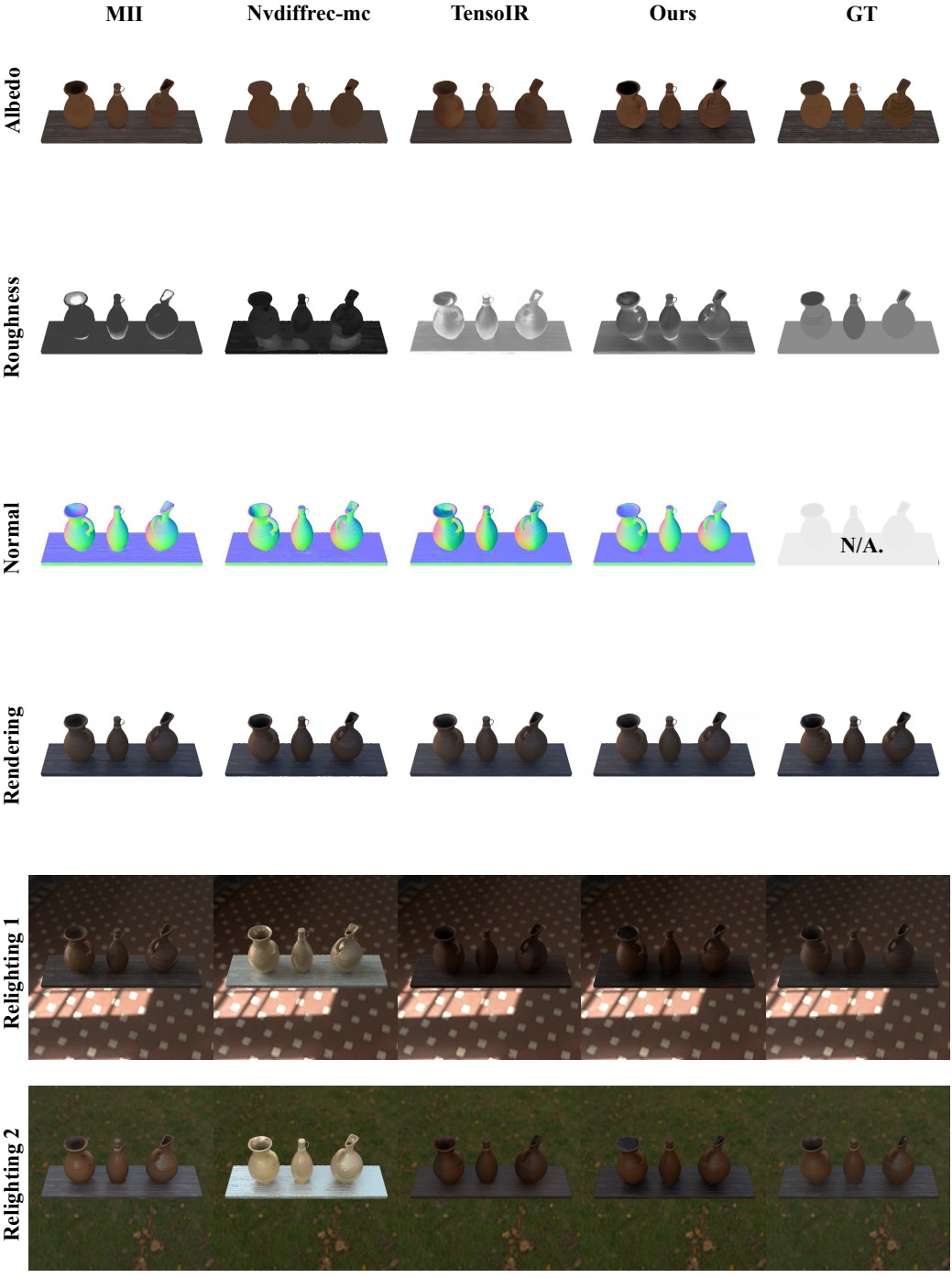

Figure 15: Qualitative comparison on *jugs* from the MII synthetic dataset.

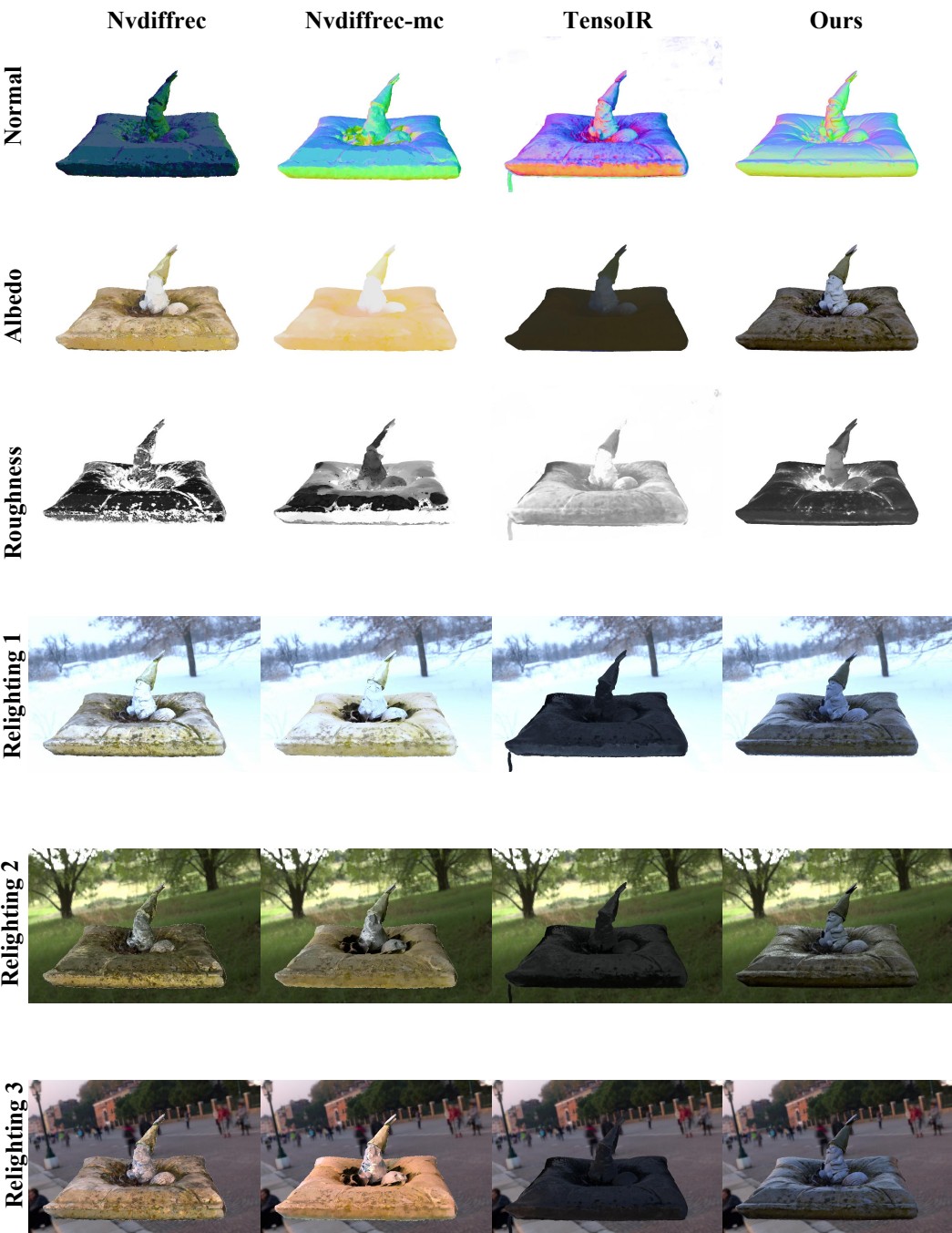

Figure 16: Qualitative comparison on *Gnome* from the NeRD dataset.

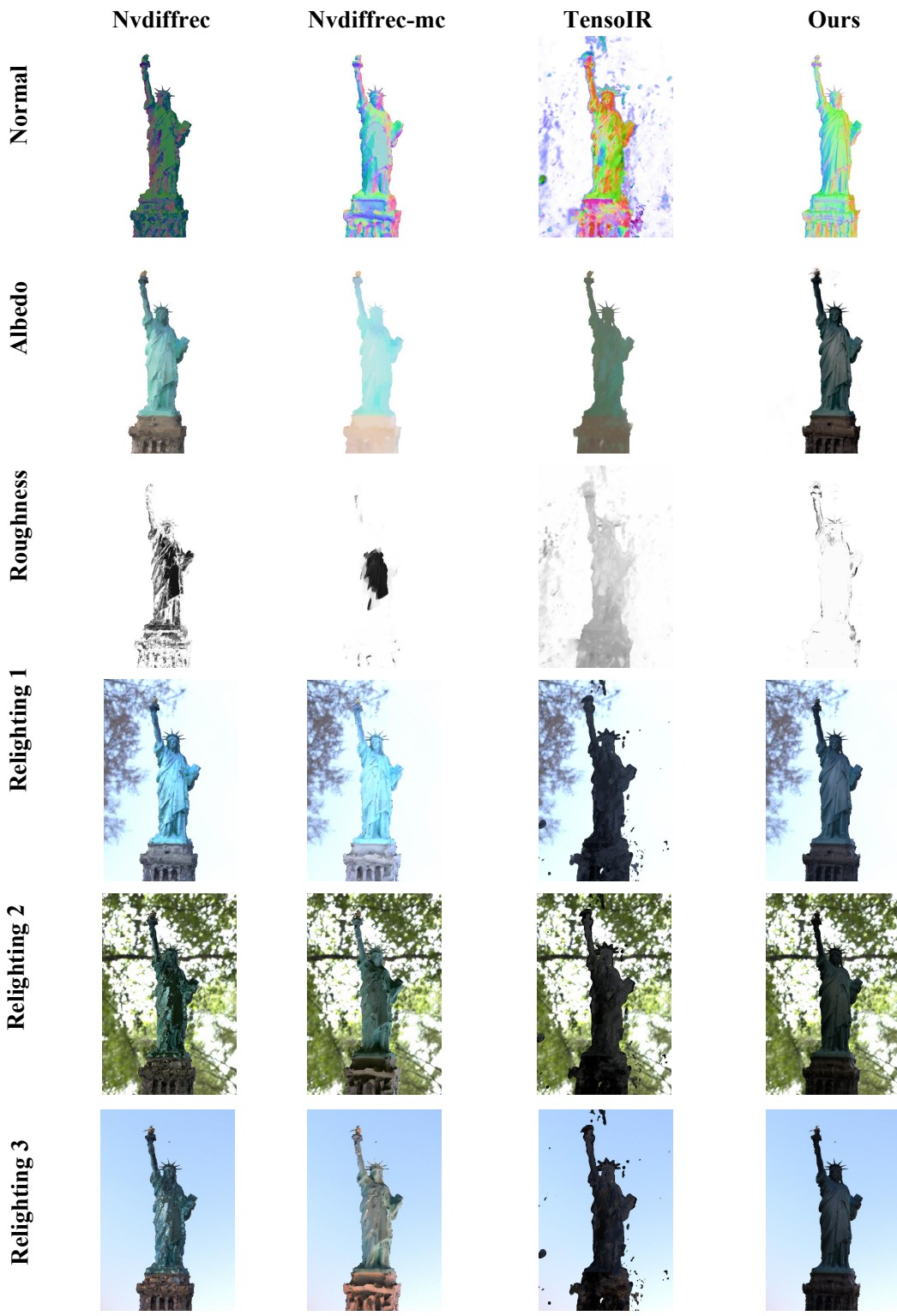

Figure 17: Qualitative comparisons on *StateOfLiberaty* from the NeRD dataset.

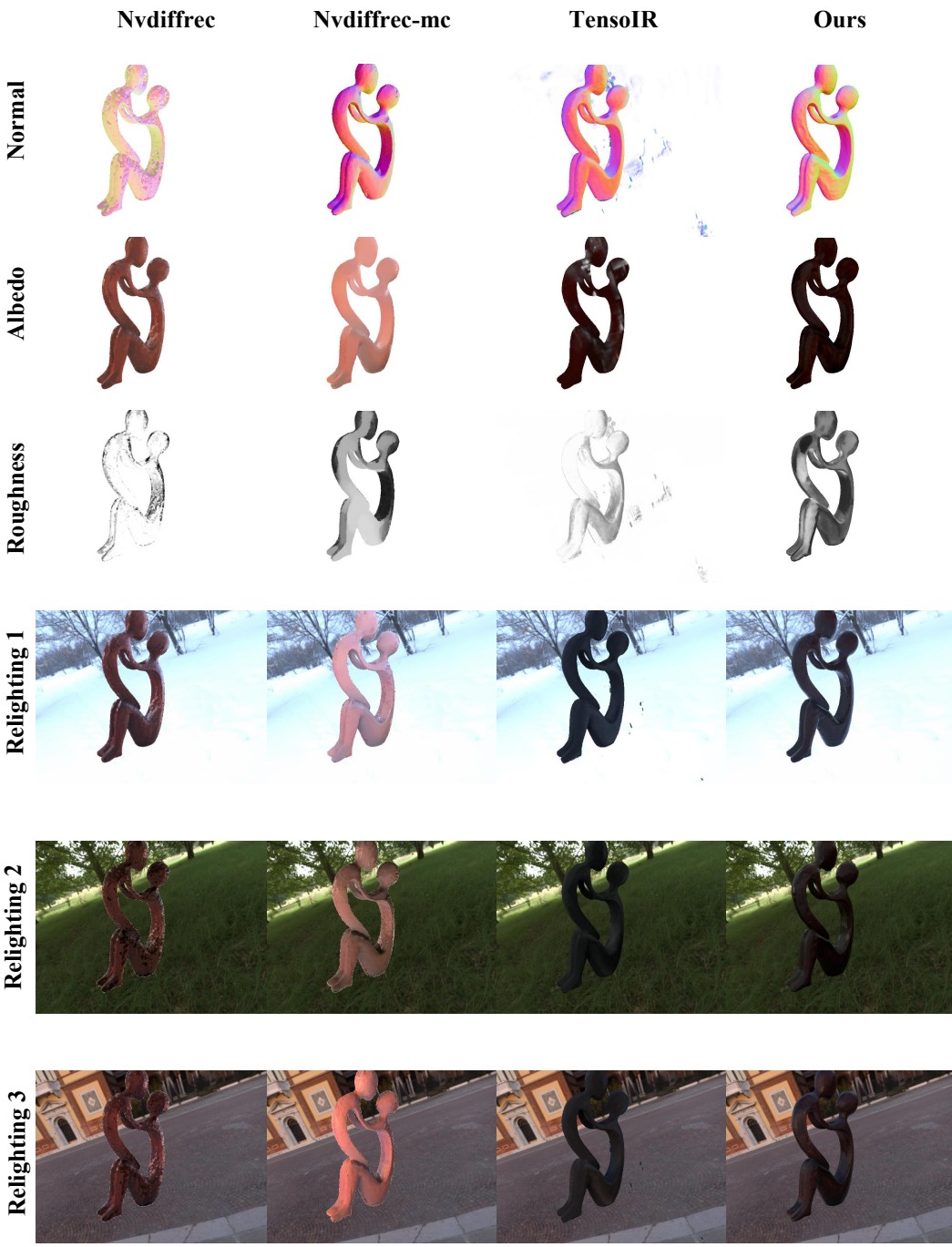

Figure 18: Qualitative comparisons on *MotherChild* from the NeRD dataset.

