# OpenReview forum: "Fast Inverse Rendering by Unified Voxelization of Scene Representation"
_ICLR.cc/2024/Conference — Submitted to ICLR 2024_

### Official Review · Reviewer_1NZF · 2023-10-13

**Soundness:** 2 fair
**Presentation:** 2 fair
**Contribution:** 2 fair
**Rating:** 5
**Confidence:** 5

**Summary:**

This paper propose a unified voxelization framework for inverse rendering (UniVoxel). UniVoxel can achieve fast inverse rendering, In addition, in order to better integrate with explicit frameworks, Spherical Gaussians to learn the incident light field was proposed.

**Strengths:**

Compared with other inverse rendering methods, UniVoxel significantly improves optimization efficiency, reducing the training time for each scene from a few hours to **18 minutes**.

**Weaknesses:**

1. The explicit scene representation of UniVoxel is very similar to Voxurf, except that it additionally predicts various properties of object materials in space. It is difficult to evaluate if adapting the semantic field during model training is novel in this paper.
2. UniVoxel cannot recover Envmap compared to other related works, which may limit its application scenarios.
3. From the experimental results, as shown in Figure 3, there is no obvious advantage in predicting the effect of materials, and the results of relighting are not very prominent. As shown in Figure 5, it seems that albedo did not decompose successfully and retained the light and dark shadows of the scene itself.

**Questions:**

1. Can you provide a video comparing the training speed with other methods? As demonstrated by the author of Plenoxels (Plenoxels vs. NeRF)  ref: https://alexyu.net/plenoxels/
2.  A question about the explicit voxelization of scene representation is that can the explicit voxelization of scene representation model reflective surfaces such as **metal or mirror** ? Will the proposed method work better on those objects than NeRF ?
3. The paper can include more related works, e.g.,
   - Yang, Wenqi, et al. "Ps-nerf: Neural inverse rendering for multi-view photometric stereo." ECCV, 2022.
   - Wang, Zian, et al. "Neural Fields meet Explicit Geometric Representations for Inverse Rendering of Urban Scenes." CVPR, 2023.
   - Mai, Alexander, et al. "Neural Microfacet Fields for Inverse Rendering." ICCV. 2023.
   - Zhang, Youjia, et al. "NeMF: Inverse Volume Rendering with Neural Microflake Field." ICCV. 2023.

---

> ### Author Response · Authors · 2023-11-17
> **Response to Reviewer 1NZF (1/2)**
>
> Thank you for your insightful feedback and thorough review of the our paper. We carefully respond to each of the concerns and questions below.
>
> **[Q1]:** The explicit scene representation of UniVoxel is very similar to Voxurf [1], except that it additionally predicts various properties of object materials in space.
> **[A1.1]:** Thanks for your valuable comment. We agree that some previous works like Voxurf have explored explicit scene representation, but this is not the main contribution of our method. We emphasize that our contributions are twofold. First, in contrast to previous methods that only use voxel grid to model geometry or appearance, we design a unified framework of scene representation, which allows for efficient learning of geometry, materials and illumination in a unified manner.
> **[A1.2]:** Second and more importantly, as also mentioned by reviewer Ft2v, we propose to leverage Spherical Gaussians (SG) to model the local incident light fields, which is capable of modeling the joint effects of direct lighting, indirect lighting and light visibility without the need of expensive multi-bounce ray tracing. The SG parameters of surface points can be predicted directly through voxel grid and lightweight MLP networks, which enables seamless integration of illumination modeling into the unified voxelization framework, thus significantly improving training efficiency.
> **[A1.3]:** We conduct experiments on Shiny Blender dataset [2] and the results are shown in **Table 4** and **Figure 9** of the appendix. We also highlight the comparison results with Voxurf in the following table, demonstrating that our method achieves better geometry quality than Voxurf.
> | Methods  | Normal MAE ↓ |
> ------------ | :----:
> | Voxurf  | 18.110 |
> | Ours | 9.292 |
>
> **[Q2]:** UniVoxel cannot recover Envmap compared to other related works, which may limit its application scenarios.
> **[A2.1]:** Thank you for your valuable comment. In theory, it is possible to filter the incident light map of all surface points to recover the environment map, but more sophisticated design is required to eliminate the influence of indirect lighting.
> **[A2.2]:** Although our UniVoxel, by utilizing SG to model the local incident light field, cannot directly acquire the environment map, it achieves comparable accuracy to the state-of-the-art methods while having higher training efficiency (about 40 $\times$ faster than MII [3] as shown in **Table 1** of the paper), which is the main advantage of our approach.
> **[A2.3]:** Environment map and incident light field are two different technical approaches for representing illumination. The former can recover the environment lighting of the scene, but the obtained lighting is not sufficiently good for direct application. On the other hand, the latter has higher training efficiency (18 minutes vs 58 minutes as shown in **Table 2** of the paper), although it cannot directly obtain the environment light. In the future, we will explore how to combine the advantages of these two methods.
>
>
> **[Q3]:** There is no obvious advantage in the experimental results.
> **[A3.1]:** Thank you for your meticulous comment. We acknowledge that our UniVoxel does not have significant advantages in terms of performance compared to other methods. However, as shown in **Figure 3** of the paper, our UniVoxel is able to recover more high-frequency details on the albedo maps. Additionally, our UniVoxel achieves comparable reconstruction quality to the state-of-the-art methods, with significantly higher training efficiency, which is the main advantage of our method.
> **[A3.2]:** Actually, we can control the shading component on the albedo maps by balancing the weight of the SG smoothness loss $L_{sg}$. In **Figure 11** of the appendix, we have demonstrated that the light and dark shadows on the albedo maps can be eliminated by reducing the weight of $L_{sg}$. Due to the complexity of lighting condition in outdoor scenes, the constraints on illumination should be relaxed to achieve more realistic materials.
>
> **[Q4]:** Can the explicit voxelization of scene representation model reflective surfaces such as metal or mirror?
> **[A4]:** Yes, we have conducted experiments on the challenging Shiny Blender dataset [2]. The quantitative results in **Table 4** of the appendix show that our UniVoxel achieves better geometry quality compared to other NeRF-like methods. The visualization in **Figure 10** of the appendix also demonstrates that our UniVoxel is capable of reconstructing more realistic materials such as metal.

---

> ### Author Response · Authors · 2023-11-17
> **Response to Reviewer 1NZF (2/2)**
>
> **[Q5]:** Can you provide a video comparing the training speed with other methods?
> **[A5]:** Yes, we have included the video in the supplementary materials. It can be observed that when the training of our UniVoxel is almost completed, MII [3] is still in the process of recovering geometry. While the normal and albedo estimated by TensoIR [4] have rough outlines, their detailed parts still need several hours to optimize.
>
> **[Q6]:** More related works.
> **[A6]:** Thanks for pointing this out. We have added the reference of these works in the related work section.
>
> **Reference:**
> [1]. Wu et al., Voxurf: Voxel-based efficient and accurate neural surface reconstruction.
> [2]. Verbin et al., Ref-NeRF: Structured View-Dependent Appearance for Neural Radiance Fields.
> [3]. Zhang et al., Modeling Indirect Illumination for Inverse Rendering.
> [4]. Jin et al., TensoIR: Tensorial Inverse Rendering.

---

### Official Review · Reviewer_iqeo · 2023-10-30

**Soundness:** 4 excellent
**Presentation:** 4 excellent
**Contribution:** 3 good
**Rating:** 5
**Confidence:** 5

**Summary:**

The paper introduces a volumetric-based inverse rendering technique notable for its efficiency. It showcases impressive experimental outcomes; however, the paper's innovation is somewhat restrained. Core aspects of the methodology appear to be previously explored in established works like PhySG, Tensorir, and NeuS. The experimental scope could benefit from an extension to include tests on the challenging Shiny Nerf-synthetic dataset, which is known for its shiny object rendering complexity. Furthermore, the paper could enhance its technical credibility by scrutinizing and refining the accuracy of its claims.

**Strengths:**

1.	The manuscript is well-composed, displaying a clear and articulate writing style.
2.	The inverse rendering outcomes presented are visually appealing and demonstrate good quality.
3.	Additionally, the related work section is comprehensive, encompassing a broad spectrum of the existing research in this field, which underscores the authors' thorough understanding of the domain.

**Weaknesses:**

1. The paper's innovative contribution is nuanced, as it applies a latent volumetric representation to enhance efficiency in inverse rendering—a concept that has been extensively studied. Although it offers incremental advancements, the overall novelty is tempered by similarities to existing methods, such as those utilizing MLPs. Additionally, the application of Spherical Gaussians (SGs) in the context of lighting is previously detailed in works like PhySG, suggesting the paper's approach is not entirely unprecedented.

2. While the paper acknowledges PhySG, it does not sufficiently recognize the prior exploration of SGs for modeling incident light. A more explicit acknowledgment would strengthen the paper by correctly attributing the origins of this idea.

3. The utilization of environment map-based lighting is a well-trodden area in research, evidenced by efforts like Nvidia’s nvdiffrec-mc. The paper's critique of environment map-based lighting in favor of SGs may not be entirely justified, and the simplifications made in visibility reasoning warrant a closer examination. The unconventional modeling of the environment map using 128 Spherical Gaussians in this paper departs from traditional methods and calls into question the asserted superiority of SG over environment maps. Therefore, claims regarding the advantages of SG should be made with greater technical precision and careful comparison.

**Questions:**

1. The paper would benefit from an explanation of the observed floating noise in TensoIR's results, providing clarity on whether this is an artifact of the algorithm, a limitation of the model, or an issue with the dataset or experimental setup used.

2. The inclusion of experiments on the challenging shiny NeRF-synthetic dataset could significantly enhance the paper's empirical foundation. Successful results on this dataset would serve as a robust testament to the algorithm's capabilities.

3. It is important for the paper to detail the distinctions between the environment map used in this study and that employed by Nvdiffrec, particularly since the paper touts the superiority of Spherical Gaussians (SG) over environment maps. A clear comparison would illustrate the specific contributions and advantages of the proposed method.

4. For a more convincing demonstration of the algorithm’s performance in handling complex lighting scenarios, the paper should present an experiment visualizing the incident light maps reconstructed using the shiny NeRF-synthetic dataset. This would showcase the practical utility of the algorithm in a real-world application, particularly for scenes with challenging lighting conditions.

---

> ### Author Response · Authors · 2023-11-17
> **Response to Reviewer iqeo (1/2)**
>
> Thank you for your insightful feedback and thorough review of the our paper. We carefully respond to each of the concerns and questions below.
>
> **[Q1]:** The novelty of the proposed method.
> **[A1.1]:** Thank you for your meticulous comment. We agree that volumetric representation has been studied in recent works, but this is not the main contribution of our method. We emphasize that our contributions are twofold. First, in contrast to previous methods that only use voxel grid to model geometry or appearance, we design a unified framework of scene representation, which allows for efficient learning of geometry, materials and illumination in a unified manner. Second and more importantly, as also mentioned by Reviewer Ft2v, we propose to leverage Spherical Gaussians (SG) to model the local incident light fields.
> **[A1.2]:** Although some works like PhySG [1] has explored the application of SG in the context of lighting, they directly represent the entire scene’s environment map with SG, requiring expensive multi-bounce ray tracing. In contrast, we utilize SG to model the local incident light radiance, which is capable of modeling the joint effects of direct lighting, indirect lighting and light visibility. And the SG parameters of surface points can be predicted directly through voxel grid and lightweight MLP networks, which enables seamless integration of illumination modeling into the unified voxelization framework, thus significantly improving training efficiency.
> **[A1.3]** We have reorganized the introduction section of the paper to clarify these major contributions.
>
>
> **[Q2]:** While the paper acknowledges PhySG, it does not sufficiently recognize the prior exploration of SGs for modeling incident light.
> **[A2]:** Thanks for your suggestion. We have added more descriptions about PhySG in section 2.1 and section 3.4 of the main paper, and explained the differences between our approach and it.
>
> **[Q3]:** The unconventional modeling of the environment map using 128 Spherical Gaussians in this paper departs from traditional methods and calls into question the asserted superiority of SG over environment maps.
> **[A3.1]:** Thanks for your valuable comment. Following your suggestion, we have adopted the conventional modeling of environment map, representing it as a learnable parameter of size 256x512x3. As shown in the following table, it obtains better reconstruction quality than using mixture of SG to represent environment map, and achieves comparable results to our method, but it also leads to much longer training time.
> **[A3.2]:** Environment map and incident light field are two different techniques for representing illumination. Methods based on environment map can usually achieve better reconstruction quality, but they require more training time due to the need of multi-bounce ray tracing. On the other hand, methods based on incident light field usually can achieve higher training efficiency. In the future, we will explore how to combine the advantages of both methods and obtain high-quality reconstruction effects with shorter training time.
> | Methods |  MVS PSNR ↑ | Albedo PSNR ↑ | Roughness MSE ↓ | Relighting PSNR ↑ |  Training Time ↓ |
>  ------------ | :----: | :----: | :----: | :----: | :----: |
> | Mixture of 128 SG  | 34.185 | 27.368 | 0.012 | 27.446 | 58 minutes |
> | 256x512x3 learnable map | 35.042 | 29.369 | 0.010 | **29.592** | 2 hours |
> | Ours  | **36.232** | **29.933** | **0.007** | 29.445 | **18 minutes** |
>
> **[Q4]:** Explanation of the observed floating noise in TensoIR's [2] results.
> **[A4]:** Thank you for your insightful suggestion. TensoIR has achieved impressive results on the MII synthetic dataset [3], but its performance in real-world scenes is poor and may have floating noise. The main reason is that the environment in each view of the real-world scenes is not completely static, making it difficult to handle complex lighting with a single environment map. One solution is to use a separate environment map for each view, but this would significantly increase training time. Our approach only requires introducing the view embedding as an additional input to the illumination model, allowing us to effectively handle complex real-world lighting without affecting training efficiency.

---

> ### Author Response · Authors · 2023-11-17
> **Response to Reviewer iqeo (2/2)**
>
> **[Q5]:** Experiments on the challenging shiny NeRF-synthetic dataset.
> **[A5.1]:** Following your suggestion, we conducted experiments on the Shiny Blender dataset [4]. The quantitative and qualitative results of the estimated normal maps are shown in **Table 4** and **Figure 9** of the appendix. It can be observed that our approach achieves better geometry quality compared to our methods, especially in the specular surfaces.
> **[A5.2]:** We also present the qualitative comparison of geometry, materials and illumination on the Shiny Blender dataset in **Figure 10** of the appendix. Although our method does not recover the details of illumination, it can be able to predict realistic albedo and roughness, while TensoIR fails to reconstruct materials in the specular surfaces and bakes the lighting into the albedo maps.
>
> **Reference:**
> [1]. Zhang et al., PhySG: Inverse Rendering with Spherical Gaussians for Physics-based Material Editing and Relighting.
> [2]. Jin et al., TensoIR: Tensorial Inverse Rendering.
> [3]. Zhang et al., Modeling Indirect Illumination for Inverse Rendering.
> [4]. Verbin et al., Ref-NeRF: Structured View-Dependent Appearance for Neural Radiance Fields.

---

### Official Review · Reviewer_qhoe · 2023-10-31

**Soundness:** 2 fair
**Presentation:** 3 good
**Contribution:** 2 fair
**Rating:** 5
**Confidence:** 4

**Summary:**

The submission proposes a 3D scene representation based on voxel grid features and MLP decoders to achieve more efficient inverse rendering. Geometrical and scene properties are encoded separately in two grid-based volumes. For scene properties (e.g., material, illumination), the volume stores implicit features in the grids, and separate MLPs are trained to decode these features into the target property values. In the case of the geometrical volume, SDF values are directly stored in the grid structure. The experimental evaluation demonstrates that this representation leads to efficient inverse rendering and delivers performance comparable to state-of-the-art methods.

**Strengths:**

+This submission proposes a unified grid-based representation that incorporates geometry and material properties. The grid-based representation has shown to be more friendly for optimization.

+Optimization becomes more straightforward with shallower MLPs. The combination of grid-based volume representation and shallow MLPs leads to more efficient optimization.

+Spherical Gaussian (SG) representation exhibits higher-order frequency characteristics for illumination compared to spherical harmonics (SH).

**Weaknesses:**

-My major concern is that the utilization of feature grids for more efficient training (the major claim) is not a novel insight. For instance, in InstantNGP, it has been demonstrated that the combination of dense hashable grids and shallower MLPs significantly enhances efficiency. Furthermore, the memory concerns highlighted in this submission can potentially be alleviated by adopting more efficient data structures for the grid, such as hashable grids as employed in InstantNGP.

-The observed performance improvement is incremental and does not surpass the performance achieved by previous methods in some cases. For instance, a notable limitation of the proposed method is that it often results in albedo estimates that include shading components as shown in Fig.5 first row, in contrast to TenSor RT.

-The assertion that SH cannot effectively model higher-frequency illumination may not be conclusive, as it is based on testing only up to order-3 SH. Increasing the order of SH could potentially address this limitation. To provide a more compelling evaluation, it is advisable to ensure an equal number of parameters when comparing SG and SH representations.

-Details missing: It is not mentioned clearly how the global illumination and self-occlusion are handled during relighting. Please see questions below for more detailed questions.

**Questions:**

1. For SG-based illumination representation, how many parameters are used for representing the Gaussians? Is it comparable to the SH counterpart?

2. How are global illumination and self-occlusion handled during relighting? During scene reconstruction, those could be baked in the SG/SH-based per-point illumination. But during relighting, the per-point environment might should be different.

---

> ### Author Response · Authors · 2023-11-17
> **Response to Reviewer qhoe**
>
> Thank you for your insightful feedback and thorough review of the our paper. We carefully respond to each of the concerns and questions below.
>
> **[Q1]:** The novelty of the utilization of feature grids for more efficient training.
> **[A1.1]** Thank you for your valuable comment. We agree that some prior works have explored the use of feature grids to improve training efficiency, but this is not the main contribution of our method. We emphasize that our contributions are twofold. First, in contrast to previous methods that only use voxel grid to model geometry or appearance, we design a unified framework of scene representation, which allows for efficient learning of geometry, materials and illumination jointly.
> **[A1.2]** Second and more importantly, as also mentioned by Reviewer Ft2v, we propose to leverage Spherical Gaussians (SG) to model the local incident light fields, which is capable of modeling the joint effects of direct lighting, indirect lighting and light visibility without the need of expensive multi-bounce ray tracing. The SG parameters of surface points can be predicted directly through voxel grid and lightweight MLP networks, which enables seamless integration of illumination modeling into the unified voxelization framework, thus significantly improving training efficiency.
> **[A1.3]** We have reorganized the introduction section of the paper to clarify these major contributions.
>
>
> **[Q2]** The memory concerns.
> **[A2]** Thanks for your construstive suggestion. The volumetric representation of our method is not in conflict with the hashable grids proposed in InstantNGP [1], and it is easy to apply it to our framework. We plan to implement it in the future.
>
> **[Q3]:** A notable limitation of the proposed method is that it often results in albedo estimates that include shading components.
> **[A3]:** Thanks for your meticulous comment. Actually, we can control the shading component on the albedo maps by balancing the weight of the SG smoothness loss $L_{sg}$. In **Figure 11** of the appendix, we have demonstrated that the shading components can be eliminated by reducing the weight of $L_{sg}$. Due to the complexity of lighting condition in outdoor scenes, the constraints on illumination should be relaxed to achieve more realistic materials.
>
> **[Q4]:** Equal numbers of parameters when comparing SG and SH representations.
> **[A4]:** Thanks for your valuable commet. Following your suggestion, we use a higher order SH representation of illumination (order-4 with 75 parameters). For a fair comparison, we also adopt the SG with 12 Gaussian lobes (72 parameters) for lighting modeling. The results on the MII synthetic dataset are presented in the following table. It can be observed that using higher order SH leads to some improvement in reconstruction performance, but using fewer parameters, SG performs better than SH, which demonstrates the effectiveness of SG in lighting modeling.
> | Methods  |  MVS PSNR ↑ | Albedo PSNR ↑ | Roughness MSE ↓ | Relighting PSNR ↑ |
>  :------------: | :----: | :----: | :----: | :----:
> | SH (order-3 with 48 parameters)  | 35.328 | 29.185 | 0.020 | 28.981 |
> | SH (order-4 with 75 parameters)  | 35.344 | 29.200 | 0.016 | 28.813 |
> | SG (12 lobes with 72 parameters)  | **36.177** | **29.746** | **0.008** | **29.148** |
>
> **[Q5]:** How are global illumination and self-occlusion handled during relighting?
> **[A5]:** Thanks for your meticulous comment. We have added the relighting procedure in the Section B of the appendix. As the incident light field obtained from previous training is not applicable to the new illumination, we adopt a similar procedure as previous works like TensoIR [2] and NerFactor [3], where we compute light visibility using Eq.7 in the main paper and consider only direct lighting.
>
> **Reference:**
> [1]. MÜLLER et al., Instant Neural Graphics Primitives with a Multiresolution Hash Encoding.
> [2]. Jin et al., TensoIR: Tensorial Inverse Rendering.
> [3]. Zhang et al., NeRFactor: Neural Factorization of Shape and Reflectance Under an Unknown Illumination.

---

### Official Review · Reviewer_Ft2v · 2023-11-02

**Soundness:** 3 good
**Presentation:** 4 excellent
**Contribution:** 3 good
**Rating:** 8
**Confidence:** 5

**Summary:**

This paper proposes a unified voxel representation to allow efficient reconstruction of geometry, material and illumination from multi-view images captured around the object. Compared with previous work that requires expensive ray tracing to compute lighting and visibility, the proposed method shows that a local spherical Gaussian illumination representation can achieve similar or even better quality of inverse rendering while significantly reducing the optimization time. Experiments on widely used real and synthetic dataset shows the overall improvements of the proposed method against state-of-the-arts.

**Strengths:**

1. Clear novelties and improvements compared to previous work.
In my opinion, the major novelty of the proposed method is a local illumination model that bakes direct illumination, visibility and indirect illumination into a spherical Gaussian representation that can be efficiently predicted. This novel representation enable significant acceleration compared to previous environment map representation, as it can avoid expensive multi-bounce ray tracing. One may expect this new method will cause more baking issue as it is less constrained compared to a global environment map lighting representation. However, the experiments show that the BRDF reconstruction quality is comparable or even better than state-of-the-arts.

2.  Comprehensive experiments
Author did comprehensive experiments on novel view synthesis, material estimation and relighting on widely-used real and synthetic datasets. Both the quantitative and qualitative results show clear improvements and the optimization time is much less compared to previous works. Authors also did ablation studies between different lighting representations, which makes the results reported in the paper more convincing.

3. Well-written paper, with all necessary implementation details included in the main paper and supplementary material.
The paper is well-written and easy to follow. With the details provided in the supplementary material, it should not be too difficult to reimplement the paper.

**Weaknesses:**

1. More focused on novelty.
Several recent methods use feature volume plus MLP to jointly reconstruct materials and geometry, such as TensorIR and NeuralPBIR. The true difference of the proposed method and previous works is the local illumination model that can accelerate the optimization. Therefore, I feel authors can emphasize this more. For example, the introduction gives me the impression that this method is faster because it uses a voxel-based scene representation, which has adapted by many previous work, but instead it probably makes more sense to emphasize the illumination model.

2. Geometry reconstruction.
I feel one experiment that is missing is the geometry quality, especially compared to TensoIR. While it is mentioned in section 4.2, it is difficult to tell the differences. I am curious how will different lighting representations impact the geometry quality, especially for the highly specular surfaces and concave regions?

3. Reference.
One con-current work that can be discussed in the paper is the NeuralPBIR (Sun et al.), which accelerates the reconstruction process by precomputing visibility and GI from NeRF so that material reconstruction can be done through local optimization. The idea of avoiding expensive ray tracing is related, while I feel the local illumination model proposed here is more different from previous works.

4. Missing details.
One details I did not find in the paper is after predicting the SG parameters, how will you render the appearance? Are you going to sample rays uniformly, with importance sampling or just compute the integral analytically? Please remind me if I miss this part in the paper.

5. Extension of the method to handle volumetric object.
One potential extension of this work is that instead of computing per-ray SG parameters and the render the appearances, we can also consider use per-point SG parameters to compute the radiance at every point through rendering equation. In this way, we might be able to reconstruct fury objects, with some modifications of the BRDF model. Does it make sense to authors?

**Questions:**

Most of my questions in the weakness section. I have two more questions.

1. About limitation in Sec. F. When you say the proposed method needs more GPU memory, how much more GPU memory does it require?
2. In Figure 13, the color of the normal maps from different methods are very different. I wonder if that is caused by any visualization issue, like different way to turn normal into RGB image?

**Details Of Ethics Concerns:**

I do not have any ethical concern.

---

> ### Author Response · Authors · 2023-11-17
> **Response to Reviewer Ft2v**
>
> Thank you for your insightful feedback and thorough review of the our paper. We carefully respond to each of the concerns and questions below.
>
> **[Q1]:** Emphasis on the illumination model.
> **[A1]:** Thanks for insightful suggestion. Following the suggestion, we have rephrased the introduction section and emphasize more on the novelty regarding the proposed illumination model.
>
> **[Q2]:** Comparison of geometry quality.
> **[A2.1]:** Thanks for your suggestion. Due to the lack of ground truth normal in MII dataset, we conducted additional experiments on the Shiny Blender dataset [1], and the quantitative results are shown in **Table 4** of the appendix. We also highlight the comparison results with TensoIR in the following table, showing that our method achieves more accurate geometry quality.
> **[A2.2]:** In **Figure 9** of the appendix, we demonstrate qualitative comparisons with other methods and different illumination models. It can be seen that our SG-based local illumination model produces higher geometry quality on specular surfaces compared to using environment maps. Additionally, using SH and NeILF as illumination models can achieve comparable geometry quality to ours. However, the quality of material recovered using our illumination model is superior to others, and the training efficiency is also higher, as demonstrated in **Table 2** of the paper.
> | Methods  | Normal MAE ↓ |
> :------------: | :----:
> | TensoIR  | 49.747 |
> | Ours | 9.292 |
>
> **[Q3]:** Reference of Neural-PBIR.
> **[A3]:** Thanks for pointing us to the interesting work. We have added the reference of Neural-PBIR in the related work section.
>
> **[Q4]:** How will you render the appearance? Are you going to sample rays uniformly, with importance sampling or just compute the integral analytically?
> **[A4]:** We discuss how to render the appearance in the implementation details (Section B) of the appendix: we utilize Fibonacci sphere sampling method over the half sphere to sample incident lights for each surface point.
>
> **[Q5]:** Using per-point SG parameters to compute the radiance at every point through rendering equation instead of computing per-ray SG parameters.
> **[A5]:** Thank you for your constructive suggestion. One challenge of using per-point SG parameters to calculate radiance is that the rendering equation needs to be calculated for each sampling point on a ray during training. Compared to rendering with per-ray SG parameters, which only require one calculation of the rendering equation per ray, this approach requires several times more GPU memory and significantly prolongs training time. But I believe this is a very insightful idea, and we will explore ways to reduce the computational cost of this method to reconstruct fury objects.
>
> **[Q6]:** How much more GPU memory does it require?
> **[A6]:** We present the batch size and required GPU memory for training each method on the MII synthetic dataset in the following table. It can be seen that our method does not significantly exceed the GPU memory of other methods, thanks to our efficient implementation. However, for larger-scale scenes, we would need to increase the resolution of voxelization and also require more GPU memory. Nevertheless, this can be addressed by the hashable grids proposed in InstantNGP [2]. And we plan to implement it in the future.
> | Methods  | batch size | GPU memory |
> :------------: | :----: | :----:
> | TensoIR [3]  | 4096 | ~12 GB  |
> | MII [4]     | 1024 | ~14 GB  |
> | Ours     | 8192 | ~19 GB  |
>
> **[Q7]:** Why are the color of the normal maps from different methods very different?
> **[A7]:** Thank you for pointing this out. The way Nvdiffrec and Nvdiffrec-mc convert normal maps into RGB images is very different from other methods. We are working on it and will address this issue in the final version.
>
> **Reference:**
> [1]. Verbin et al., Ref-NeRF: Structured View-Dependent Appearance for Neural Radiance Fields.
> [2]. MÜLLER et al., Instant Neural Graphics Primitives with a Multiresolution Hash Encoding.
> [3]. Jin et al., TensoIR: Tensorial Inverse Rendering.
> [4]. Zhang et al., Modeling Indirect Illumination for Inverse Rendering.

---

> ### Comment · Reviewer_Ft2v · 2023-11-22
>
> Thanks for the answers! I think my questions to the paper have been resolved.

---

> ### Author Response · Authors · 2023-11-23
> **Official Comment by Authors**
>
> Dear Reviewer Ft2v,
>
> Thank you for taking the time and effort to review our response! If you have any further questions, please let us know and we will respond promptly!
>
> Thank you once again for your time and attention.
>
> Best,
>
> Paper 2483 Authors

---

### Author Response · Authors · 2023-11-21
**Response overview**

We thank all the reviewers for their constructive and insightful suggestions. We are glad to see that the reviewers find that:
1. Our method has clear novelties compared to previous work (Reviewer Ft2v).
2. The proposed framework achieves favorable reconstruction quality while significantly improving the optimization efficiency (Reviewer Ft2v, qhoe and 1NZF).
3. The experiments are comprehensive and the results are convincing (Reviewer Ft2v).
4. Our paper is well-written (Reviewer Ft2v and iqeo3), displaying a clear and articulate writing style (Reviewer iqeo3).

We have responded to each reviewer’s comments diligently and incorporated their valuable suggestions (all modifications are highlighted in blue), as summarized in the following:
1. We reorganize the introduction of the paper to clarify the major novelty of our method in Section 1: Although some previous works have explored explicit representations, this is not the main contribution of our method. We emphasize that our contributions are twofold. First, in contrast to previous methods that only use voxel grid to model geometry or appearance, we design a unified framework of scene representation, which allows for efficient learning of geometry, materials and illumination jointly. Second, we propose to leverage Spherical Gaussians (SG) to model the local incident light fields, which is capable of modeling the joint effects of direct lighting, indirect lighting and light visibility without the need of expensive multi-bounce ray tracing. The SG parameters of surface points can be predicted directly through voxel grid and lightweight MLP networks, which enables seamless integration of illumination modeling into the unified framework, thus significantly improving training efficiency (concern of Reviewer Ft2v, qhoe, iqeo3 and 1NZF).
2. We discuss more con-current works in Section 2 (concern of Reviewer Ft2v and 1NZF).
3. We discuss the differences between our approach and other methods like PhySG in illumination modeling in Section 3.4 (concern of Reviewer iqeo).
4. We add the experimental results on the Shiny Blender dataset in Section D, Table 4, Figure 9 and Figure 10 of the appendix (concern of Reviewer Ft2v, iqeo3 and 1NZF).
5. We add the experiments to investigate the effects of illumination constraints on the shading components of the albedo map in Section E and Figure 11 of the appendix (concern of Reviewer qhoe and 1NZF).
6. We add the video comparing the training speed with other methods in the supplementary materials (concern of Reviewer 1NZF).
7. We add the relighting procedure in Section B of the appendix (concern of Reviewer qhoe).

We have addressed the other concerns raised by each reviewer in the corresponding official comments.
Finally, we would like to express our sincere gratitude to the reviewers and remain open to further improvements in all aspects of our work.

---

### Meta-Review · Area_Chair_SYj6 · 2023-12-05

**Metareview:**

The paper proposes a 3D scene representation based on unified voxel grid features and MLP decoders to achieve efficient inverse rendering. Compared to previous work that requires expensive ray tracing, the proposed method shows that a local spherical Gaussian illumination representation can achieve equally good quality inverse rendering with significantly reduced computation cost. Experiments show favorable quality and computation performance,

The major strengths of the paper are:
(1) A unified grid-based representation that incorporates geometry and material properties that is friendly for optimization.
(2) The combination of the grid-based representation and MLP decoders leads to efficient optimization.
(3) The inverse rendering results are visually appealing.

On the other hand, the chief weakness is that the latent volumetric representation proposed in the paper is conceptually not new, but it has been extensively studied. In addition, since the focus of the paper is on efficiency, the thread of research works in the same spirit, e.g., InstantNGP that uses dense hashable grids, should be fully discussed.

Overall, while the reviewers appreciate the work and quality results, there remains a concern about the novelty of the work.  The reviewers and AC read the rebuttal and took it into consideration to reach the final recommendation.

**Justification For Why Not Higher Score:**

Although we see the effectiveness of the proposed method in terms of quality and computation performance, the main claim of the paper about the novel unified grid-based representation (i.e., latent volumetric representation) is not fully justified. In addition, approaches such as InstantNGP in a similar spirit in pursuing efficiency, should be thoroughly discussed.

**Justification For Why Not Lower Score:**

N/A

---

### Decision · Program_Chairs · 2024-01-16

Reject